# Hypomethylating Chemotherapeutic Agents as Therapy for Myelodysplastic Syndromes and Prevention of Acute Myeloid Leukemia

**DOI:** 10.3390/ph14070641

**Published:** 2021-07-04

**Authors:** Vincent G. Sorrentino, Srijan Thota, Edward A. Gonzalez, Pranela Rameshwar, Victor T. Chang, Jean-Pierre Etchegaray

**Affiliations:** 1Department of Biological Sciences, Rutgers University—Newark, Newark, NJ 07102, USA; vincent.sorrentino@rutgers.edu (V.G.S.); st981@scarletmail.rutgers.edu (S.T.); egonzalez1796@gmail.com (E.A.G.); 2Department of Medicine, Division of Hematology/Oncology, Rutgers New Jersey Medical School, Newark, NJ 07103, USA; rameshwa@njms.rutgers.edu; 3Veteran Affairs New Jersey Health Care System, East Orange, NJ 07018, USA; Victor.Chang@va.gov

**Keywords:** epigenetic, myelodysplastic syndrome, acute myeloid leukemia, DNA methylation, decitabine, azacytidine, bortezomib, cancer, hypomethylating agent, myeloma

## Abstract

Myelodysplastic Syndromes (MDSs) affect the elderly and can progress to Acute Myeloid Leukemia (AML). Epigenetic alterations including DNA methylation and chromatin modification may contribute to the initiation and progression of these malignancies. DNA hypomethylating agents such as decitabine and azacitidine are used as therapeutic treatments and have shown to promote expression of genes involved in tumor suppression, apoptosis, and immune response. Another anti-cancer drug, the proteasome inhibitor bortezomib, is used as a chemotherapeutic treatment for multiple myeloma (MM). Phase III clinical trials of decitabine and azacitidine used alone and in combination with other chemotherapeutics demonstrated their capacity to treat hematological malignancies and prolong the survival of MDS and AML patients. Although phase III clinical trials examining bortezomib’s role in MDS and AML patients are limited, its underlying mechanisms in MM highlight its potential as a chemotherapeutic for such malignancies. Further research is needed to better understand how the epigenetic mechanisms mediated by these chemotherapeutic agents and their targeted gene networks are associated with the development and progression of MDS into AML. This review discusses the mechanisms by which decitabine, azacitidine, and bortezomib alter epigenetic programs and their results from phase III clinical trials.

## 1. Introduction

Myelodysplastic syndrome (MDS) is a hematological malignancy characterized by aberrant hematopoiesis and most commonly affects the elderly [1]. From 2007 to 2011, the reported incidence of MDS per 100,000 individuals in the United States was 4.9 per year; today, the number is likely higher due to increased awareness of the condition [2]. MDS patients display cytopenia, leading to a susceptibility for infection and bleeding [3]. Principal diagnostic criteria for MDS include the presence of dysplasias in the peripheral blood and bone marrow; in both tissues, more than one cell lineage can be affected [3]. MDS subtypes are defined by cytopenia type and cell lineage-specific dysplasias. For example, RARS or MDS-RS is defined by refractory anemia, dysplasia associated with erythropoiesis, the presence of ringed sideroblasts (RSBs), and erythroid precursor cells with abnormal iron accumulation in the mitochondria around the nucleus [4]. Differential blood tests are helpful in prognostic classification of dysplasias and hematopoietic insufficiencies; higher blast counts in the bone marrow (5–10%) and low peripheral blood cell counts are often associated with more severe form of the disease [3]. The 2016 World Health Organization classification of myelodysplastic syndromes and neoplasms recognizes categories of MDS with single lineage dysplasia, MDS with multilineage dysplasia, MDS with ringed sideroblasts, MDS with excess blasts, and MDS Unclassifiable [5]. The IPSS-R stratifies patients at diagnosis into risk groups based upon cytogenetics, marrow blast proportion, hemoglobin, absolute neutrophil count, and platelet count [6]. High risk MDS patients can progress to acute myeloid leukemia (AML) [1]. AML is a hematological cancer characterized by immature myeloid cell proliferation with blasts >20% and often accompanied by bone marrow failure [7]. Cytological features of the bone marrow such as Auer bodies, crystalline rod-like structures found in the cytoplasm of leukemic myeloid cells, are common in AML and can be helpful in diagnosis of the disease [8].

The last decade of research has characterized various epigenetic and genetic abnormalities linked to both MDS and AML, resulting in impaired genetic machinery and hematopoietic stem cell function, all of which are more common in the aged population [1,7]. More than 30 driver mutations have been identified in MDS patients; these mutations affect DNA methylation pathways, RNA splicing, chromatin modifications, transcription, and signal transduction [9]. Non-random chromosomal mutations have been identified as genetic events that promote disease initiation and progression and are common in approximately 52% of AML patients [10]. Somatic mutations of genes that encode epigenetic, transcription factors and signaling proteins in AML patients [11], include DNA methyltransferase 3A (DNMT3A), Ten-Eleven Translocation 2 (TET2), CCAAT Enhancer Binding Protein α (CEPBA), p53 and tyrosine kinase receptors [7]. This review selects three specific epigenetic drugs capable of inhibiting DNA methylation: decitabine, azacitidine, and bortezomib.

Decitabine was first synthesized in 1964 [12] as a cytostatic compound and then identified as a suitable treatment for leukemia in murine strains with distinct MHC-II class in 1968 [13]. Decitabine (5-aza-2′-deoxycytidine) has been the focus of numerous studies to better understand the epigenetic basis for cancer treatments. Early experimentation pointed to decitabine’s anti-leukemic properties in mouse models. A 1978, leukemic mice injected with decitabine with an 8-h infusion showed an increase in lifespan with little to no toxicity [14]. Additionally, in 1983, treatment of leukemic mice with longer infusion of decitabine resulted in long-term survivors [15]. These early studies formed the basis for the use of decitabine as a chemotherapy for leukemia.

Studies on decitabine at lower doses uncovered its specific inhibitory effects on DNA methylation, and thereby identified as a hypomethylating agent (HMA) [16]. One of the initial studies found decitabine and azacytidine, another cytidine-analog, to induced gene expression via DNA demethylation (Figure 1A–C) [16]. Subsequent research confirmed decitabine as a DNA methyltransferase inhibitor (DNMTi) that targets 14,000 regulatory regions of genes in different cancer cell types derived from lung [17], prostate [18], colon [19] and blood cancers [20]. An important study reported the use of decitabine at three low doses in patients diagnosed with high-risk MDS and chronic myelomonocytic leukemia (CML). The study found the best results with a dose of 20 mg/m^2^ intravenously for 5 days [21]. In 2006, the U.S. Food and Drug Administration (FDA) approved decitabine as a treatment for MDS and CML [22].

Azacitidine (5-azacytidine), an analog of cytidine with similar structure to decitabine, acts as a DNMT inhibitor by modifying the 5th carbon of the pyrimidine ring (Figure 1A). Instead of a carbon atom bonded to a hydrogen atom, azacitidine consists of a nitrogen atom in the same position [23]. The analog was first synthesized in 1963 by Pίskala and Šorm at the Institute of Organic Chemistry and Biochemistry in Prague [24,25]. Soon after, the azanucleoside was isolated from the gram-positive bacteria *Strepotoverticillium ladakanum*, and its functionality with respect to cytotoxicity and anti-proliferation, were shown [26]. After initial discovery and synthesis, azacitidine was used for its anti-metabolite properties in disturbing normal metabolic processes, leading to its use as a chemotherapeutic for leukemia [23]. In 1971, the nucleoside analog was used for its cytostatic properties in chemotherapy, as it proved efficacious in the treatment of childhood leukemia. In the 1980s, the demethylating properties of the drug were identified. This led to further studies and clinical trials to investigate the drug as an epigenetic modulator [25]. With the linkage between DNA hypermethylation and the development of cancer, azacitidine, as a hypomethylating agent, was quickly sought after for its antineoplastic properties, specifically through reactivation of previously silenced genes, including tumor-suppressors [23,27].

In 2004, nearly 40 years after its discovery, Azacitidine (Vidaza; Celgene), an injectable suspension, was approved by the FDA, followed by approval in 2008 by the European Medicines Agency (EMA) [23,27]. Vidaza, a bioavailable formulation of 100 mg of azacitidine and 100 mg mannitol, has been used for subcutaneous administration in MDS, AML and CMML patients [23]. Currently, the hypomethylating agents azacitidine and decitabine, are the recommended therapeutic treatment for MDS patients classified by the International Prognostic Scoring System (IPSS) at intermediate-2/high risk [28]. Subcutaneous injections of azacitidine, however, require the need for in-person treatments and potentially cause injection site infections; this can be detrimental for AML patients with severe alterations in bone marrow blasts. Recently, azacitidine has become available as an oral formulation CC-486 (ONUREG; Celgene), which is being clinically tested in patients with MDS, AML, and CMML [29,30]. This version was approved by the FDA in 2020 as a maintenance treatment for AML patients who failed intensive induction chemotherapy and achieved a first complete remission due to the successful trials in the QUAZAR AML-001 study (ClinicalTrials.gov Identifier: NCT01757535) on 1 September 2020 [31].

In addition to hypermethylation, aberrant proteolysis has been associated with the development of particular malignancies; as such, targeting protein degradation processes has been the focus of many researchers interested in developing anticancer drugs [32]. One of the crucial pathways involved in protein degradation is the ubiquitin-proteasome pathway (UPP), which is responsible for targeting degradation of approximately 80% of cellular proteins [33]. This pathway is executed by the proteasome: a multiprotein complex that is responsible for the recognition and degradation of ubiquitin-marked proteins [33]. Malignant cells tend to accumulate defective proteins due to their increased synthesis capability; this increases their reliance on the proteasome’s disposal mechanism [34,35]. Therefore, any compound that interferes with proteasomal disposal of proteins could result in the accumulation of defective proteins, increased cellular stress, and apoptosis of malignant cells [35]. Since the proteasome degrades and processes mediators of the cell-cycle and apoptosis such as cyclins, caspases and nuclear factor of kB (NF-kB) [35], this molecule also drives the cell cycle via regulation of protein levels that activate and/or inhibit phase transitions in cell growth and replication [36]. Moreover, several studies have demonstrated proteasome inhibition results in cell death [33] due to impaired degradation of p53 [37] and p21 [38].

The chemotherapeutic effect of bortezomib relies on its ability to regulate protein turnover. Bortezomib is a proteasome inhibitor known to induce apoptosis in malignant hematopoietic cells [35,39]. First synthesized by Myogenics in 1995, bortezomib showed early in vitro and in vivo results in clinical studies [32] and was the first proteasome inhibitor to be used clinically in the treatment of malignancies, namely refractory/relapsed myelomas [40]. After extensive clinical trials, it was approved by the FDA in 2003 for the treatment of relapsed multiple myeloma [32,41]. Current research investigates bortezomib-mediated mechanisms associated with epigenetic pathways implicated in cancer development and progression [42].

Determining the mechanisms and targeted genes by decitabine, azacytidine and bortezomib remain an active area of research in the context of hematological malignancies and other cancers. The purpose of this review is to underline relevant findings with respect to each of these drugs in MDS and AML. The identification of new epigenetic biomarkers linked to these drugs, in the context of MDS and AML, can be used for developing new targeted therapies to increase efficacy of existing treatments.

## 2. Main Body of Review

### 2.1. Decitabine (5-Aza-2′-Deoxycytidine)

#### 2.1.1. Mechanism of Action

Decitabine is a cytidine analog where nitrogen replaces the carbon five in the pyrimidine ring (Figure 1B,D). In general, decitabine represents a DNA hypomethylating agent with two main mechanisms of action (Figure 2). At low doses, decitabine reactivates silenced genes and promotes cellular differentiation. At high doses, it elicits cytotoxic effects, leading to cell death [16,22]. In general, azanucleosides like decitabine are chemically unstable and considerations must be taken when identifying safe and effective methods to administer such drugs [43]. Decitabine and its metabolites bind more tightly to enzymes that mediate its incorporation into DNA when compared to the natural substrate; for example, the drug binds 10 times more tightly to cytidine deaminase than deoxycytidine [44].

Like many drugs, the effectiveness of decitabine treatment is dependent upon the targeted cells to transport the drug [43]. Mechanistically, decitabine is incorporated intracellularly by nucleoside transport proteins, including the equilibrative uniporters (ENTs; SLC29A family) and concentrative uptake transporters (CNTs; SLC28A family), which have both been directly linked to the uptake of chemotherapeutic analogues in the treatment of leukemias [43]. The drug is then targeted by deoxycytidine kinase, which converts decitabine (5-aza-dCR) to 5-aza-2′-deoxycytidine monophosphate (5-aza-dCMP) (Figure 1D) [43]. Through a series of phosphorylation reactions by nucleoside di-phosphokinase [44,45], 5-aza-dCMP is further converted into the active nucleotide form 5-aza-2′-deoxycytidine-5′-triphosphate (5-aza-dCTP), which can substitute for cytosine during DNA replication and thereby incorporated into DNA (Figure 1D) [43,46]. DNA methyltransferase enzymes will then recognize 5-aza-dCTP-guanine dinucleotides for DNA methylation (Figure 2) [43]. DNMT1 becomes inactivated by forming an irreversible covalent bond with the 5-aza-dCTP-guanine dinucleotide (Figure 2) [47]. This covalent bond results in a rapid loss of DNMT activity as bound enzymes are unable to carry out further downstream methylation activity [48], leading to global hypomethylation (Figure 2) [45]. This process is known as enzyme DNA adduct formation. Specifically targeting the S-phase of the cell cycle [44], decitabine leads to antitumor action via inhibition of proper DNA replication in cancerous cells; however, the drug also interferes with transcription and DNA repair processes (Figure 2) [49].

Experimental evidence suggests that the effectiveness of low-dose decitabine in malignancies stems from its ability to demethylate silenced tumor-suppressor genes related to leukemic malignancies and other myeloid disorders [49] such as p15^INK4b^, E-cadherin and *MYOD1* [50,51]. Moreover, various distinct methylation patterns have been identified in AML patients, some predictive of clinical outcomes [20]. As mentioned above, nucleoside uptake is an important process in the effectiveness of decitabine treatment regimens. Reduced expression of uptake transporters like SLC22A4 have been reported as strong predictors of poor event-free and overall survival in AML patients [52]. DNA methylation-based epigenetic repression could be a contributing factor to such poor outcomes, as pre-treatment with decitabine restored SLC22A mRNA expression, increased cellular uptake of anthracyclines, and was associated with increased sensitivity to cytarabine, a chemotherapeutic, in human AML cell lines [52].

Decitabine has also been shown to induce the expression of tumor-associated antigens, resulting in induced immune cytotoxic effect, indicating its indirect role in immune therapy [49]. MDS patients treated with 1.3 µM decitabine at a dose of 15 mg/m^2^/day showed improved expression of cancer-testis antigens (CTAs; i.e., MAGE-A1, MAGE-A3, and SP17) against solid tumors [53]. Decitabine treatments were accompanied by enhanced T-lymphocyte recognition of MDS cells, specifically in response to increased CTA expression [53]. This study also found decitabine treatments in MDS patients to increase T-lymphocyte function, expression of HLA class antigen and ICAM-1, a cell adhesion and co-stimulatory molecule in adaptive immunity responses [53,54]. This underlines opportunities for decitabine treatment regimens to be used in-tandem with immunotherapies already in use and highlights the wide variety of genetic targets affected by decitabine’s mechanism of action.

#### 2.1.2. Decitabine Hematological Malignancies

Hypomethylating agents are attractive because of their reduced toxicity in elderly patients when compared to standard induction chemotherapy [55]. As mentioned above, anti-tumor activity of decitabine in hematological malignancies results from multiple mechanisms including, induction tumor-suppressor genes upon hypomethylation, enzyme DNA adduct formation, activation of apoptotic pathways and induction of tumor-associated antigens [49,53]. In AML cell lines such as OCI-AML2, decitabine was found to induce the expression of 81 genes while inhibiting the expression of 96 genes; whereas, normal peripheral blood cells displayed significantly fewer changes in gene expression [56]. Interestingly, this study showed that nearly 50% of decitabine-induced genes are deprived of CpG methylation sites in their promoter regions, which suggest a decitabine-mediated effect that is independent of DNA methylation status.

In patients with hematological malignancies like MDS and AML, decitabine is at its most effective when used over a prolonged period with fractionated exposures [57,58]. Moreover, response rates to decitabine tend to be slow, with most patients requiring at least two to five monthly cycles to achieve maximum clinical results [59]. DNA demethylation in response to decitabine is a function of the dose, in which lower doses (0.1 µM) induced higher levels of DNA hypomethylation, whereas very low (0.01 µM) and high (10 µM) decitabine doses induced significantly lower DNA hypomethylation with concomitant higher rate of cytotoxicity. A 2017 study examined the effect of low dose (20 mg/m^2^/day × 5 days) decitabine treatments in lower risk MDS patients. The results show 70% of patients displayed an overall response rate (ORR) to decitabine treatments and 32% of these patients became blood transfusion-independent [60], demonstrating an overall hematological improvement. One trial interested in dose/schedule dependent responses to the drug enrolled MDS patients in one of two distinct treatment regiments: 3-day treatments (3 h IV infusion of 15 mg/m^2^ given every 8 h for 3 consecutive days every 6 weeks) or 5-day treatments (1 h IV infusion of 20 mg/m^2^ given once daily on days 1–5, every 4 weeks; [61]). Overall Response Rate for the 3-day group was 29.4% and 25.5% for the 5-day group with the median AML-free survival time was 23.8 months, and 24-month overall survival was 48.9% [61].

Recent phase III clinical trials have demonstrated varied success with decitabine treatments. These trials typically use criteria like progression-free survival (PFS) defined as duration from the date of treatment to progressive disease or death, event-free survival (EFS) defined as the time from study entry until relapse/malignancy/death, and overall response rate (ORR) defined as the percentage of participants who achieved stringent complete response, very good partial response, or partial response, to quantify the effectiveness of the drug therapy in question. In a study comparing decitabine (15 mg/m^2^ every 8 h for 3 days) with best supportive care (BSC) in elderly patients (>60 years) with MDS, 15% of those receiving decitabine showed complete/partial remission and hematological improvement; patients receiving decitabine also had longer PFS than those receiving BSC, but similar overall survival (OS) in both groups [62]. Another phase III randomized study looking at decitabine in individuals with MDS found an overall improvement in patient outcomes when treated with the drug. Patients received either 15 mg/m^2^ over 3 h every 8 h decitabine IV repeated every 6 weeks, or BSC; decitabine patients showed an ORR of 17%, with 9% complete responses, compared to 0% response for BSC group [63]. Decitabine responses were classified as durable (median, 10.3 months) and prolonged patients’ median time to AML progression when compared to BSC patients [63].

Additionally, researchers have been searching for improved biomarkers of predictive of clinical outcomes in MDS/AML patients undergoing decitabine treatment. A trial in patients with MDS who have unfavorable-risk cytogenetic profiles and TP53 mutations, have shown increased response rates to decitabine in MDS treatment protocols [19,59]; therefore, higher-risk individuals may be more sensitive to the drug. Another trial looking at high-risk MDS patients with varying cytogenetic profiles found decitabine responses specific to patients’ cytogenetic profile, namely regarding autosomal monosomies (MK−, MK+, MK1, MK2+) [64]. ORR in cytogenetically normal (CN) patients was 36.1%, 16.7% in MK− patients, and 43.6% in MK+ patients; PFS was prolonged in CN and MK2+ groups but not MK−/MK+/MK1 subgroups when compared to BSC patients [64]. One in vivo study found elevated fetal hemoglobin (HbF) to be a reliable predictor of overall survival in MDS/AML patients, with decitabine elevating HbF levels in 81% of MDS patients and 54% of AML patients [65]. These studies underline the importance of patient karyotype when deciding which treatment regimens may be most effective in MDS/AML treatment and what physiological signals clinicians should assess for reliable predictors of patient outcome.

#### 2.1.3. Side Effects and Complications

During decitabine’s initial development phase in the mid-1980s, toxicities associated with the drug’s use included myelosuppression, nausea, and mild to moderate fatigue [16]. Contemporary studies report neutropenia, febrile neutropenia, thrombocytopenia, leukopenia, anemia, fatigue, nausea, diarrhea, and constipation [66]. In general, one of the main concerns with the use of decitabine is its cytotoxic effects at high doses [45], especially at non-target sites. Despite this concern, decitabine and azacitidine tend to have less cytotoxic effects compared to other more conventional chemotherapeutics; this is because the dosage needed to elicit desirable epigenetic effects is relatively low compared to the dosage that elicits cytotoxic effects [16]. Possible myelosuppressive complications like infection/bleeding, or prolonged myelosuppression [defined as hypocellular marrow] have also been reported but tend to be less common than the milder adverse side effects mentioned above [21].

Despite success with decitabine in the treatment of hematological malignancies, some MDS/AML patients do not respond effectively to the drug [67]. Many patients fail to respond to initial decitabine treatments (primary resistance); those that do respond initially often relapse and become unresponsive to subsequent exposures (secondary resistance) [68,69]. Moreover, treatment with decitabine is not curative; different forms of resistance to the drug have been identified over the years [70], most notably due to alterations in the pathways that activate and metabolize decitabine, as well as downstream mutations of genes involved in DNA methylation/demethylation dynamics such as TET2 [71], DNMT3A, and ASXL1 [72]. Thus, a better understanding of these contributing pathways can help researchers and clinicians identify which mechanisms should be targeted at different stages of MDS and AML progression to develop salvage therapies.

Researchers have begun looking for ways to bolster the effects of decitabine using combination therapies with other drugs. Polo-like kinases regulate the cell cycle [73], various portions of mitosis, and contributes to DNA damage repair and replication stress [74]; therefore, these proteins make a reasonable target for anticancer therapeutics. Volasertib and rigosertib, two polo-like kinase inhibitors currently under phase III trial review, are currently being used in combination with decitabine in the treatment of MDS/AML patients who are ineligible for intensive remission therapy [75]. 3-Dezazneplanocin (DZNep), a histone methyltransferase inhibitor, has also been identified as a drug that could bolster decitabine’s efficacy in MDS/AML patients [67]. Together, decitabine and DZNep have demonstrated synergistic activation of several tumor suppressor genes and synergistic activation of apoptosis in human AML cell lines HL-60 and AML-30 [67]. These types of studies widen options for developing new combination treatment regimens.

### 2.2. Azacitidine

#### 2.2.1. Mechanism of Action

Similar to decitabine, azacitidine elicits two distinct properties, cytotoxicity and DNA hypomethylation, depending on dosage schedules. At high dosage, azacitidine promotes cytotoxicity due to its incorporation into both RNA and DNA, while low dosage prompts DNA hypomethylation effects (Figure 1D and Figure 2D) [76]. The anti-proliferation effect on abnormally dividing hematopoietic cells in the S-phase of the cell cycle results from interference with nucleic acid metabolism [77]. Thereby, further nucleic acid synthesis and cellular proliferation are impaired, and apoptotic pathways are activated.

Azacitidine’s effect is initiated after intake of the oral CC-486 or injection by Vidaza via cellular uptake carried out by specific transmembrane proteins; the nucleoside analog is transported into a cell via a human concentrative nucleoside transport 1 (hCNT1) which is part of the SLC28 gene family of three subtypes that transport naturally occurring nucleosides and synthetically engineered nucleoside analogs [23]. Upon transport into the cell, azacytidine is phosphorylated by uridine cytidine kinase to 5-azacitidine 5′-triphosphate, its active conformation [23]. The compound is then degraded or integrated into the nucleic acids. Triphosphate nucleosides are degraded in the cells by cytidine deaminase (CDA) and converted to 5-azauridine compounds, rendering them inactive [23]. Concentration and synthesis of CDA in human liver and spleen influences the half-life of the drug, making it approximately 41 min in vivo. Moreover, reduced levels of active drug by degradation decreases the drug’s efficacy and potency. The various inactive metabolites of azacitidine are primarily discarded via urine secretion with minimal (<1%) fecal excretion [78].

Azacitidine is incorporated into the genome of rapidly proliferating cells during the S-phase [23,79] and does not show sufficient affinity for non-proliferating cells [80]. Upon integration into RNA, azacitidine interrupts RNA metabolism and translation processes. Furthermore, azacitidine can also incorporate into DNA by ribonucleotide reductase-mediated conversion of azacitidine diphosphate into 5-aza-2′-deoxycytidine diphosphate. Phosphorylation of the diphosphate into triphosphate enables azacitidine to be integrated into DNA during replication [27]. Azacitidine has an increased affinity to RNA over DNA with a ratio of incorporation of 65:35, respectively, in AML cell lines [76]. Azacytidine-integrated genomic regions cause DNMTs to form an irreversible complex with DNA [23]. Specifically, DNMTs (3a and 3b) interact with azanucleosides as if they are natural nucleosides by forming a covalent bond with carbon-6 of the pyrimidine ring. This bond is irreversible due to azacytidine’s nitrogen atom that replaces carbon 5 of cytosine, which prevents the β-elimination reaction that allows dissociation of interacting DNMTs (Figure 1 and Figure 2). This complex remains intact until DNMTs are eventually degraded [47,81]. This results in gradual loss of CpG island methylation and epigenetic memory during cellular proliferation (Figure 2) [23]. Collectively, azacitidine induces hypomethylation of DNA by inhibiting DNMTs and reactivating previously hypermethylated genomic regions, resulting in altered genomic expression patterns.

#### 2.2.2. Azacitidine in Hematological Malignancies

Patients with MDS display increased hypermethylation around regions of the DNA associated with regulating tumor-suppression and normal cell proliferation in tumor cells. Methylation changes in the 9p21 chromosomal regions consisting of the cyclin-dependent kinase inhibitor 2B (*CDKN2B*) and cyclin-dependent kinase inhibitor 2A (*CDKN2A*) genes have been frequently associated with myelodysplastic syndromes and AML. Both genes are involved in the inactivation of cyclin dependent kinases to mediate over the cell cycle G1 progression [82]. Aberrant methylation of the *CDKN2B* gene, which encodes for the tumor suppressor p15^INK4b^, is frequently hypermethylated and silenced [83,84]. In MDS subtypes, hypermethylated regions have been identified in the *CDKN2A* gene in 38% and *CDKN2B* gene in 77% of patients. Increased degree of methylation was noticed in AML patients of 77% in *CDKN2A* gene and 100% in *CDKN2B* gene [82]. As such, the level of methylation is directly correlated with disease progression. Patients suffering from RAEB and RAEB-T (5–20% abnormal blasts in bone marrow) [85] embodied increased risk of consisting hypermethylated regions in p15^INK4b^ than patients with RA and RARS (<5% abnormal blasts in bone marrow) due to the apparent differences in disease progression and developed bone marrow blasts. DNA methyltransferase inhibitor, 5-azacitidine, demonstrated significant hypomethylation and reduced mRNA expression of *CDKN2B, IGSF4*, and *ESR1* genes followed with hematological improvements in MDS patients [86]. Moreover, another study demonstrated an average *CDKN2B* hypomethylation of 6.8% in 47% of the patients with azacitidine, [87] which corroborated with the suppression of *CDKN2B* and tumor growth in SKM-1 cell lines in vitro [88].

Other studies confirmed the following silenced genes to be significantly reactivated by azacitidine and decitabine: tissue inhibitor of metalloproteinase 3 (*TIMP3*), *p16*, cyclin-dependent kinase inhibitor 1C (*CDKN1C*), and RAS association domain family 1 (*RASSF1*) [23]. Overall, DNMTs 3a and 3b cause the reactivation of genes involved in apoptosis, cell cycle, and DNA repair mechanisms. In an in vivo study, azacitidine reactivates the hypermethylated region of the phosphoinositide-phospholipase C (*PI-PLCβ1*) gene, which is involved in intercellular signal transduction, cell cycle modulation, proliferation, and differentiation [80,89]. Furthermore, high-risk MDS patients displayed a mono-allelic cryptic deletion of the *PI-PLCβ1* gene, and the expression patterns of the two alternative splicing isoforms of *PI-PLCβ1*, *PI-PLCβ1a* and *PI-PLCβ1b* mRNAs, were altered [90]. Azacitidine treatment corresponded with a progressive increase in the levels of *PI-PLCβ1* mRNAs until complete remission was observed for the high-risk MDS cases. The lipid signaling pathway changes due to the expression of the *PI-PLCβ1* gene and yields various levels of activated Akt; this is potentially involved in deregulation of the cell cycle and mortality of the malignant cells in MDS [90]. Accumulation of reactivated gene products along with induced cytotoxicity help regulate basic cellular mechanisms and benefit patients with hematological malignancies.

In a comparative meta-analysis study between the hypomethylating agents decitabine and azacitidine both drugs demonstrated significant incidences of complete remission for high-risk myelodysplastic syndrome patients when compared to best supportive care (BSC) [28]. However, neither drug displayed significant difference in terms of mortality rates, hematological improvements, and patient responsiveness in the meta-analysis conducted by Almasri et al. (2018) [28]. Recommend dosage for subcutaneous injectable azacitidine is 75 mg/m^2^ during the first treatment cycle, but the dose may be increased to 100 mg/m^2^ if the initial two cycles are deemed ineffective. Normal treatment schedule consists of continuous dosage for 7 days every 4 weeks. Following the recommended dosage, the CALGB 8421, CALGB 8921, CALGB 9221 phase I/II/III, respectively, studies demonstrated azacitidine provides significant response rates, improved quality of life, and reduced risk of further leukemia development when compared to BSC [91,92,93,94]. CC-486 is currently being investigated in patients with lower risk MDS, low/intermediate-1 risk MDS, and AML to serve as an alternative treatment option to the injectable form [30]. Premise for the investigations of CC-486 are due to promising factors in consistent dosing, reduced side effects, and favorable patient response when compared to Vidaza [91]. The azacitidine tablets, CC-486, are to be administered 300 mg orally once daily along with a antiemetic on days 1 through 14 of each 28-day cycle after the first complete remission is achieved.

Earlier studies show the induction of fetal hemoglobin by azacytidine as one of the first examples of epigenetic modulation. This led to the current treatments for sickle cell disease by increasing fetal hemoglobin. In this case, high levels of fetal hemoglobin suggest more successful methylation changes [95]. Pathogenesis of AML involves overexpression of the anti-apoptotic protein B-cell lymphoma 2 (BCL2), which promotes proliferation of lymphocytes. Increased BCL2 production is linked to high mortality rates in AML patients and poor responsiveness to chemotherapy [96,97,98]. As such, DiNardo et al. (2020) investigated a drug therapy combination of azacitidine and venetoclax, a BCL2 inhibitor, in a phase 3, multicenter, randomized, double-blind, placebo-controlled trial [99]. Findings exemplified the complete remission to be approximately 66.4% among the subject group who received azacitidine plus venetoclax compared to a 28.3% for the control group who solely received azacitidine. Moreover, the incidence of composite complete remission within the genomic risk subgroups (adverse cytogenetic risk, secondary AML, and high molecular mutations cohort) also proved significantly [99]. Therefore, the investigation led by DiNardo et al. (2020) proved the combination of azacitibine and venetoclax as significantly effective in improving complete remission and overall survivability. Dombret et al. (2015) [100] evaluated azacitidine against conventional care regimens (CCR) (standard induction chemotherapy, low-dose ara-c, or supportive care only) in a phase 3 trial with 488 patients age ≥65 years diagnosed with AML (>30% blasts). HMA treatment significantly increased the median overall survival by 3.8 months and one year survival rate by 12.3% when compared to CCR [100]. The studies demonstrated a potential alternative for patients who are ineligible for chemotherapy or stem cell therapy. Even after effective chemotherapy, majority of patients experience a relapse after acquiring a remission event. As such, Grövdal et al. (2010) investigated the use of subcutaneous azacitidine as a means of maintenance therapy following the primary treatment. Results in patients with either MDS, secondary AML, or CMML demonstrated shorter periods of overall survival and complete remission [101].

#### 2.2.3. Side Effects and Complications

Azacitidine is highly unstable in aqueous solutions due to an electron deficiency in position 6 of the triazine ring. Formic acid and carbamoylguanidine are spontaneously formed after a nucleophilic attack to the imine group instead of a carbon atom [25]. This can hinder the pharmacological potential of azacitidine; structural modifications (HPMP-5azaC 31 and phenyl,2-,3-,4-pyridinyl [102]) to 5-azacytosine ring either decreased the function or deemed the molecule biologically inactive. Formulations of azacitidine (Vidaza and CC-486) are based on a myriad of clinical and laboratory trials of the drug’s efficacy in patients with hematological malignancy subtypes. Most common adverse reactions associated with the DNMT inhibitor treatment are thrombocytopenia, neutropenia and leukopenia (grade 3/4), gastrointestinal events including nausea, vomiting, and injection site reactions [91,101]. However, febrile neutropenia, anemia, neutropenic sepsis, and pneumonia were noticed in a minimal percentage of patients in the AZA PH GL 2003 CL 001 clinical study. Myelosuppression and infections are common forms of toxicity experienced by patients with subcutaneous azacitidine treatment [103]. Moreover, subcutaneous administration of Vidaza leads to post-injection site erythema, ecchymosis, and inflammation, potentially causing swelling, itching, pain, redness, warmth, rash, or hives [91]. Injection site injuries generally demonstrate mild symptoms unless prolonged for extended duration or worsen in condition. Treatment consists of the use of ice packs to reduce swelling and medications to manage the inflammation, pain, and potential infection. Under appropriate dosage schedules, azacitidine is a well-tolerated drug with manageable adverse side effects, allowing most patients to recover [91].

In vitro experiments demonstrated that continuous drug exposure resulted in a subsequent increase in DNA demethylation and reactivation of silenced genes. However, prolonged exposure to azacitidine will cause genetic amplification and insertional activation of oncogenic loci due to continuous inhibition of DNA methylation [43,104,105]. Higher inhibitory dose concentrations of hypomethylating agents exerted unwanted cytotoxicity, interference with DNA replication and genome damage with less effectiveness than that of lower dose concentrations [23]. Reinforcing the claim, higher doses of azacitidine as a starting dose were associated with elevated incidental occurrences of grade 3/4 neutropenia [101]. Moreover, any of the side-effects noticed with azacitidine are less severe and of shorter duration than the ones experienced from CCR.

Resistance to azacitidine is naturally developed and poses a threat to its efficacy as a therapeutic treatment for MDS and AML. Currently, mechanism associated with azacitidine resistance remain unclear. However, a mechanism described for azacytidine resistance include its deficient cellular uptake due to decreased expression of human nucleoside transporters (hNTs) [106,107]. Additionally, mutations altering the levels of deoxycytidine kinase (DCK) can promote decitabine-resistant in leukemia cells [69]. Altered expression levels of any of the enzymes involved in azacitidine phosphorylation, such as UCK, NMPK, and NDPK could potentially induce azacitidine resistance. Additionally, azacitidine resistance could result from increased expression of anti-apoptotic factors, such as BCL2L10 [108]. Upregulated BCL2L10 levels are associated with azacitidine-resistant monocytic cell lines derived from AML patients including THP-1/ARE, HL60/AR, and SKM1R cells [107,108]. Further mechanisms for azacitidine resistance in leukemia involve chromatin reorganization by the RNA methyltransferase NSUN3 and the DNA methyltransferase DNMT2, which can alter the recruitment of specific transcription factors [109]. Collectively, individual or combination of the aforementioned mechanisms could contribute to the overall acquisition of resistance against azacitidine.

Genetic screenings in monocytic cell lines show that decrease expression of DCK (deoxycytidine kinase), UCK2 (uridine cytidine kinase 2), and SLC29A1 (solute carrier family 29 member 1—Augustine blood group) genes due to accumulated mutations correlated with a resistant effect towards azacitidine and guadecitabine, a ‘next-generation’ dinucleotide decitabine analog [110]. This drug-resistance in monocytic cells was confirmed in vivo using immune-compromised mouse models displaying partial or complete resistance to azacitidine and guadecitabine. Another study found mutations in exons 4 and 5 of the UCK2 gene in THP-1 and HL60 azacitidine-resistant cells [107]. Azacitidine resistance could be attributed to an impaired inhibition of DNMT activity due to decreased UCK2 leading to reduced levels of the phosphorylated active form of azacytidine (5-aza-dCTP). In addition, this study clarified that UCK2 gene mutations, not increased BCL2L10 levels, resulted in azacitidine resistance since siRNA-mediated knockdown of BCL2L10 did not restore azacitidine sensitivity [107]. Notably, patients with MDS, AML, and CMML refractory to azacitidine and decitabine treatment displayed downregulation of UCK2 and SLC29A1 after initial relapse [111]. Resistance to azacitidine remains as a clinically significant problem.

Guadecitabine (SGI-110) is a novel, second-generation DNA methylation inhibitor that is designed to diminish the outcomings of azacitidine and decitabine and improve the efficacy and safety compared to first-generation HMAs [112,113]. Primarily, the intent of the DNMTi is to reduce the instability associated with azacitidine and decitabine since the drug is resistant to CDA. Guadecitabine is linked with a phosphodiester bond between decitabine and deoxyguanosine [112]. In an open-label, multicenter, phase 1 dose escalation study (ClinicalTrials.gov Identifier: NCT01261312), the recommended clinical and biologically effective dosage schedule for patients with intermediate or high risk-MDS or AML was 60 mg/m^2^/day × 5 [114]. The results were reinforced in a phase 1/2 study comparing 5 days versus 10 days schedule for two different dosages: 60 mg/m^2^ and 90 mg/m^2^. Percent of patients achieving complete response were similar for all drug doses and schedules of guadecitabine, but the dose schedule 60 mg/m^2^/day × 5 was deemed recommended for the patient population (median age of 77 years) [115]. The researchers of Astex Pharmaceuticals further investigated the novel HMA in phase 3 ASTRAL-2 (ClinicalTrials.gov Identifier: NCT02920008) and ASTRAL-3 (ClinicalTrials.gov Identifier: NCT02907359) trials by comparing it against physician’s alternative treatment option. Evaluation of the primary end point did not yield a statistically significant improvement in the overall survival of the patients with MDS, AML, and CMML compared to alternative therapy. Currently, studies for the secondary end points are ongoing.

### 2.3. Bortezomib (Velcade)

#### 2.3.1. Mechanism of Action

Over the years, researchers have worked to better understand the binding mechanisms of bortezomib and how it works to effectively inhibit the proteasome. To carry out protein degradation via the UPP, cells employ a complex enzymatic system that marks protein substrates with a poly-ubiquitin chain; in eukaryotic cells these marked substrates undergo proteolysis via the 26S proteasome [116]. The eukaryotic 26S proteasome consists of three important structural elements. Two 19S regulatory subunits recognize protein substrates and facilitate their entry into the catalytic 20S core particle [33]. The 20S core particle functions as a catalytic core that directly carries out the process of protein degradation. This core consists of seven β subunits, each with its own unique active site to carry out the proteasome’s enzymatic action.

Structurally, bortezomib is a dipeptide boronic acid analogue that contains pyrazinoic acid, phenylalanine, and leucine (Figure 3A) [32]. The chemical make-up of bortezomib capitalizes on specific amino acid interactions within β-subunits of the proteasome’s 20S catalytic core (Figure 3B). Bortezomib preferentially targets the β5 active site inside the 20S catalytic core of the proteasome; the boronic acid subgroup of bortezomib binds to the Thr1 residue on the β5 subunit, inhibiting the subunit’s chymotrypsin-like activity [32,38,116]. Via hydrogen bonding, Gly47, Ala49/50, and Thr21 residues within the β5 subunit stabilize the interaction between Thr1 and the bortezomib’s boronic acid subgroup [116]. Bortezomib’s boronic acid moiety also binds to amino acid residues in the β1 subunit of the 20S catalytic core, partially inhibiting this subunit’s trypsin-like activity and further contributing to the drug’s action as a proteasome blocker (Figure 3B) [42].

This proteasome inhibitory action is exemplified in bortezomib’s effect on multiple myeloma (MM) cells. MM cells overproduce immunoglobulin and cytokine proteins to help maintain contact with bone marrow stroma and facilitate their survival; this adhesion-related survival mechanism is known as cell adhesion-mediated drug resistance and is a quintessential characteristic of MM cells. However, this overproduction can result in inappropriate accumulation of misfolded and functional proteins in the ER and cytosol; therefore, MM cells rely heavily on protein degradation systems to compensate for increased ER activity and stress [34,35]. Loss of proteasome activity via bortezomib leads to an accumulation of misfolded and functional proteins, and MM cells are left uniquely susceptible to severe ER stress; this results in ER overload, excess oxygen build up, and protein dysregulation [117] leading to DNA damage and malignant cell death. Bortezomib also inhibits the expression of cellular receptors like IL-6 in MM cells [118], and decreases the expression of cell adhesion molecules on the MM cell surface [119]. This disrupts MM cells’ ability to acquire survival benefits from bone marrow stroma, making them more susceptible to standard chemotherapy treatments [34].

Bortezomib’s inhibition of the proteasome also targets genetic pathways related to malignancy specific to multiple myeloma. Bortezomib treatments have led to downregulation of transcripts associated with cellular growth and survival pathways, as well as the upregulation of transcripts associated with pro-apoptotic pathways [120]. A known pathway affected by bortezomib involves nuclear factor of kB (NF-kB) (Figure 3C) [32]. When in its active state, NF-kB is translocated into the nucleus and bind promoter regions to stimulate transcription of genes expressing growth factors/signaling molecules (IL-6, TGFb, TNFa, IGF-1, SDF-1, HGF), cell-adhesion molecules (VLA-4, VLA-5, ICAM), angiogenesis factors (VEGEFs, angioprotein-1, MCP-1), and anti-apoptotic enzymes (Bcl-2, Bcl-_XL_, A1, cIAP, XIAP, FLIP) (Figure 3C) [35]. Because of its role in promoting cell growth and preventing apoptosis, and its frequent activation in MM via mutations in NF-kB cascade components, NF-kB signaling is a reasonable target for [multiple myeloma] chemotherapeutics [121]. NF-kB becomes active when its inhibitory partner IkB is ubiquitinated by the UPP and subsequently degraded by the proteasome [122]. Bortezomib prevents the degradation of the inhibitory molecule IkB, thereby keeping NF-kB in an inactive state [123]. Consequently, bortezomib decreases the expression of anti-apoptotic proteins, leading to increased apoptosis in cancer cells (Figure 3C) [42]. In addition to promoting apoptosis via the NF-kB signaling pathway, research has shown bortezomib-induced NF-kB inactivity could also affect DNA methylation levels. Typically, the zinc finger-containing transcription factor Sp1 forms a complex with NF-kB and activates transcription of the DNMT1 gene (Figure 3C) [124]. However, with increased NF-kB inactivation, DNMT1 activity could be decreased, resulting in hypomethylation of tumor suppressors such as PDLIM4 (PDZ and LIM domain 4) gene [125].

Bortezomib has also been shown to downregulate the expression of several proteins involved in the protective cellular response to genotoxic stress [126]. Genotoxic stress like DNA damage can result in loss of apoptotic or cell-cycle arrest abilities, allowing malignant cells to form and replicate. Further DNA damage from genotoxic stress, specifically mutations in tumor suppressor and/or oncogenes, can result in particularly resistant cancer phenotypes [127]. As cancer cells generally undergo more stress and are more reliant on stress adaptive pathways, however, bortezomib’s blocking of stress responsive pathways is important in allowing chemotherapeutics to effectively damage DNA and initiate cancer cell death [128]. While bortezomib treatments increase the sensitivity of multiple myeloma cells to conventional chemotherapeutics by inducing changes in cell-adhesion molecules at the mRNA level [119], the drug also induces changes transcripts involved with regulation of cell growth, apoptosis, and heat shock response. For example, topoisomerase II beta, which relaxes DNA torsion during replication, was effectively inhibited when bortezomib was used in-tandem with conventional chemotherapeutics like mitoxantrone, doxorubicin, and etoposide [129]. Bortezomib also inhibited effectors involved in DNA base-excision repair, such as 8-oxoguanine DNA glycosylase and uracil-DNA glycosylase [130], and effectors involved in DNA mismatch repair [126].

More recently, researchers have uncovered connections between bortezomib and its effect on the epigenome, specifically RNA interference mechanisms. A study looking to uncover new molecular mechanisms of bortezomib found 719 genes and 28 miRNAs downregulated, and 319 genes and 61 miRNAs upregulated, in neuroblastoma cells treated with the drug [131]. MicroRNAs (miRNAs) can contribute to translational inhibition and mRNA degradation of specific targets [132]. Bortezomib was found to activate CEBPD (CCAAT/enhancer binding protein delta), a transcription factor associated with cellular differentiation, motility and cell death, thereby functioning as a tumor suppressor in various cancers by promoting apoptosis and cell cycle arrest [133]. Bortezomib-mediated activation of CEBPD promotes the expression of miR744, miR3154, and miR3162. These miRNAs form an inhibitory complex with Ago2 (Argonaute RISC catalytic component 2), targeting downregulation of the oncogenes PRKDC, MCM4, and UBE2V2 [134].

The mechanism by which bortezomib activates CEBPD remains unclear. CEBPD is involved in regulating hematopoietic tissue homeostasis, as several members of the CEBP family are expressed during myeloid development [135]. As a tumor suppressor, CEBPD is inactivated in multiple cancer types, including leukemia [136]. In AML, CEBPD is frequently silenced by hypermethylation [135]. CEBPD was found to be highly expressed in cells treated with drugs known to alter the epigenome (i.e., 5-AzadC), induce differentiation (i.e., retinoic acid), and stimulate the mitogen-activated protein kinase (MAPK) p38 (i.e., cisplatin, paclitaxel, 5-fluorouracil, bortezomib and dexamethasone) [136,137]. MAP kinases (MAPKs) are involved in modulating angiogenesis, proliferation, metastasis, and invasion of tumors [138]. Specifically, p38 is a stress activated MAPK that can modulate multiple cellular processes including transcription, translation, cell surface receptor expression and cytoskeletal structure, which can in turn regulate apoptotic response [139,140]. Given its role in regulating apoptosis and its biochemical function as a kinase, p38 could have a role in the bortezomib-mediated activation of CEBPD. Overall, despite research suggesting bortezomib’s effect on transcripts involved in cellular stress responses and its role in initiating RNAi, researchers are currently trying to uncover specific mechanisms by which bortezomib elicits these effects on the epigenome.

#### 2.3.2. Bortezomib and Hematological Malignancies

Cell culture studies have found bortezomib can induce apoptosis in both hematological cancers and solid tumor malignancies [35], including pancreatic [141], ovarian, and prostate [142] cancers in addition to its effects in multiple myeloma cancer cells [143]. Bortezomib has also been shown to suppress MDS/AML cell survival via proteasome inhibition [144] and acts as an indirect transcriptional inhibitor for several genes associated with AML [58]. Relatedly, the drug has been shown to induce proteasome-independent degradation of TRAF6 protein, but not mRNAs related to this protein [144]; specific degradation of this protein coincides with drug-induced autophagy and targeted cell death in MDS/AML cells [144]. The drug also enhanced the efficacy of volasertib mitotic arrest in AML cells, effectively preventing mitotic slippage and initiating malignant cell death [145].

Bortezomib has shown promise in vivo as well; in xenotransplant mouse models of human AML, mice receiving bortezomib and volasertib displayed prolonged survival and better disease control than mice receiving volasertib alone [145]. Bortezomib has been shown to induce an overall improved erythroid response in patients suffering from MDS and related hematological conditions. In a patient suffering from 5q– syndrome, a subtype of MDS, bortezomib treatment resulted in the normalization of platelet counts and beneficial alterations in thrombopoiesis cytokines [40]. In humans, phase I clinical trials demonstrated proteasome activity could be successfully inhibited via bortezomib treatment in refractory multiple myeloma patients, with dose- and time-dependent responses [32]. Phase II trials demonstrated increased success of the drug in multiple myeloma patients, especially when used in combination with other chemotherapeutics [32]. One study reported an 88% overall response rate (complete response + partial response) in untreated multiple myeloma patients treated with bortezomib and dexamethasone [146].

Most recent phase III clinical trials involving bortezomib and hematological malignancies examine the drug’s effect in multiple myeloma patients, specifically when used in tandem with other chemotherapeutics, steroids, and antibody treatments. Of the several phase III clinical trials underway, only a few currently have results published and/or reported. One phase III trial, completed and published in February 2013, sought to compare the effect of Velcade (bortezomib), melphalan and prednisone in patients with previously untreated multiple myeloma; this trial was particularly interested in exploring secondary malignancy risk in MM patients post-treatment. Patients received nine 6-week cycles of VMP (1.3 mg/m^2^ of bortezomib per day on days 1, 4, 8, 11, 22, 25, 29, and 32 on cycles 1–4 and days 1, 8, 22, 29 during cycles 5–9, alongside melphalan 9 mg/m^2^ and prednisone 60 mg/m^2^ per day of days 1–4 each cycle), or MP alone [147,148]. After 5 years, patients receiving VMP had a 31% reduced risk of death versus patients receiving only MP treatments. OS benefit was most notable in patients 75 years or older, those with stage III multiple myeloma, and creatine clearances < 60 mL/min. All three patients’ subgroups showed higher OS when treated with VMP versus MP [147]. Additionally, time to next therapy (TTNT) was markedly longer in patients receiving VMP (30.7 months) when compared to those receiving MP (20.5 months) [147]. These results demonstrate that when used in tandem with other therapies, bortezomib can increase the efficacy of multiple myeloma treatments and suggests the drug does not confer any additional risk in developing secondary malignancies post-treatment.

Another trial completed in April 2021 explores a novel treatment regimen for acute myeloid leukemia in individuals under the age of 30. Patients received the same chemotherapeutic backbone, with or without 1.3 mg/m^2^ of bortezomib given on days 1, 4, and 8 of each chemotherapy dose [149]. However, bortezomib did not improve EFS (44.8% vs. 47.0%) or overall survival (63.6% vs. 67.2%) when compared to the control group; moreover, the addition of bortezomib treatments to standard AML chemotherapy regimens resulted in significantly more peripheral neuropathy and intensive care unit admissions [149]. Overall, the results of this study do not support bortezomib’s use alongside other chemotherapies in AML treatment, as it the drug increased chemotherapy-related toxicity but did not improve survival in any significant way.

#### 2.3.3. Side Effects and Complications

Contemporary studies using bortezomib have documented several systemic side effects, the majority of which manifested as a general lack of energy and weakness in patients [41]; these include thrombocytopenia, neutropenia, diarrhea, nausea, fatigue, and anemia [41,150]. However, more recent studies have reported non-hematological toxicity of the drug, specifically with respect to its effect on the neuromuscular and cardiovascular systems [151]. A 2005 study looking at the drug’s effect in multiple myeloma found a small proportion of patients experienced peripheral neuropathy, hypercalcemia, and spinal cord compression [41].

Of these side effects, peripheral neuropathy presents a particularly difficult challenge in bortezomib treatment regimens. Bortezomib-induced peripheral neuropathy is a neurological condition characterized by mild/moderate distal sensory loss, mild to severe distal pain, mold motor weakness in distal muscles, and rare autonomic failure [152]. Researchers have found pathological evidence of this adverse effect in rats, in which bortezomib treatments induced damage to the Schwann cells and myelin of the animals’ sciatic nerves and dorsal root ganglia [153]. Bortezomib has also been reported to have several adverse effects on the cardiovascular system including the onset of arrhythmias [154], in more mild cases, and sudden heart failure in more severe cases [154,155].

Resistance to bortezomib has also become a recent clinical issue within the past decade [156]. Despite the drug’s marked success in treating multiple myeloma, up to half of patients possess some intrinsic resistance to the proteasome inhibition initiated by the drug [32]. Some patients also display acquired resistance to bortezomib, due to mutations in the β5 subunit of the 20S catalytic core [156,157] and overexpression of the β5 subunit gene [158]. Modeling of proteasomal subunit mutations in MM cell lines indicated two somatic mutations in the subunit β type 5 (PSMB5) gene (i.e., T21A and A49V) that altered their binding pocket causing a reduced binding affinity for bortezomib (Figure 4) [159]. This finding was validated in patient-derived MM cells, as both amino acid substitutions resulted in bortezomib resistant cells [159]. However, the heterogeneity of MM creates complications when attempting to address drug resistance. Initially, multiple MM patients show robust responses to bortezomib and other proteasome inhibitors; however, these patients could be heterogeneous for PSMB5 mutations causing an advantage through normal proteasomal function with the added benefit of less effective drug binding [159].

Current research has begun investigating ways to combat resistance to the drug [150]. One study looking specifically at patients who had displayed resistance to bortezomib found some success in combining drug treatments to overcome the associated resistance. Researchers found an overall response rate of 73.3% relapsed/refractory multiple myeloma patients treated with Panobinostat and bortezomib in an expansion phase cohort [150]. Of those individuals, bortezomib-resistant patients showed an overall response rate of 26.3% [150], demonstrating it was possible to address this specific complication.

## 3. Discussion

Recent clinical trials using decitabine in the treatment of hematological malignancies have determined effective dosage and dose schedules for decitabine to be promising in improving the prognosis of patients with MDS, AML, and related cancers [57,58,59]. ORR and PFS of patients suffering from these conditions have been improved with decitabine treatment regimens when compared to patients receiving best supportive care [61,62,63]. These trials have also determined genetic and physiological biomarkers predictive of patient outcome when treated with decitabine [64]. Concerns with the drug’s efficacy in patients who failed to respond to initial HMA therapy also remain an important avenue for future research when considering future clinical trials [67].

Mutations in DNMT3a are common biomarkers in AML patients at a frequency of 22–33% [160,161]. Although DNMT3b mutations are rare in AML, its high expression levels are associated with poor prognosis [161]. DNMT1 has been a focal point for mechanistic studies Involving decitabine and bortezomib in hematological malignancies, nevertheless mechanisms regarding DNMT3a and DNMT3b activity remain largely unclear upon treatment with these drugs [161]. Therefore, while DNMT1 serves as a reliable target for drugs used to treat of MDS and AML, DNMT3a mutations can offer a reliable diagnostic tool when evaluating patient prognosis and disease severity [161]. Current research is also exploring the possible development of DNMT3a inhibitors, which could provide new insights into AML development and offer a preventative treatment strategy for patients suffering from MDS and other hematological pre-cancers [162].

With incoming clinical data discussing decitabine and other HMAs’ role in inducing hypomethylation to treat pre-cancers like MDS and other hematological malignancies, research is still examining possible consistencies in demethylation. Do decitabine (and other HMAs) induce global hypomethylation? Or do they target specific gene promoters unique to the development and progression of MDS, hematological malignancies, and other cancers? A 2011 study looking for such consistencies found that decitabine and azacitidine induced non-random demethylation patterns at specific loci in human colon cancer (HCT116) and leukemic cell lines (HL60) [19]. Additionally, this study found a significant amount of overlap in the genes targeted by each drug [19]. Identifying consistent patterns in demethylation could be crucial in more targeted clinical applications of these drugs and understanding where specifically to look for their in vivo epigenetic effect.

It is already known that decitabine demethylates tumor-suppressor genes such as p15^INK4b^, E-cadherin and *MYOD1* [50,51], making it efficacious in treating various malignancies in which these genes play a crucial role. However, with frequent relapse and progression to AML in MDS patients treated with decitabine [67,70], it may be necessary to further explore the [epi]genetic roots of these drug resistances. Research has identified mutations related to decitabine resistance, specifically in genes affecting DNA methylation (TET2, [71]; DNMT3A, and ASXL1, [72]) and mutations in uptake transporters responsible for cellular uptake of azanucleosides (Figure 4) [52]. Identifying such mutations in patients before prescribing a treatment regimen could help clinicians in deciding which combination therapy can be the most efficacious for the individual patient. Moreover, direct comparison of methylation patterns distinct to MDS/AML patients and known genetic targets of decitabine may aid in developing personalized treatment regimen based on the patient’s unique cytogenetic profile.

Although mutations in epigenetic factors like DNMT1/3a and TET2 are common in hematological malignancies, there are mixed findings when it comes to whether such genetic alterations are principal contributors to secondary resistances [71,163,164,165]. For example, patients suffering from myeloid malignancies displayed effective silencing of DNMT1 in response to initial decitabine/azacytidine treatments; however, these levels rebounded during patient relapse. Therefore, secondary resistance is not necessarily rooted in DNMT1 mutation, as this would have likely appeared in initial drug treatments. These findings suggest secondary resistances, which might be associated with adaptive rather somatic mutations related to MDS and AML disease development [166]. For instance, changes in pyrimidine metabolism was suggested as an adaptive mechanism by which patients with relapsed myeloid malignancies develop secondary resistances. Gene expression profiles in relapsed patients display shifts in key pyrimidine metabolism enzymes 72–96 h post-treatment; these shifts are an attempt to preserve nucleotide levels and protect cells from acute nucleotide imbalances caused by pyrimidine analogs like decitabine and azacytidine.

Recent success with cedazuridine, an oral formulation of decitabine, points to more accessible HMA treatments in the future [66]. Within the past year, a fixed dose oral combination (FDC) of decitabine and cedazuridine has been developed by Astex pharmaceuticals and is currently being examined in various clinical studies [120]. With orally bioavailable HMAs on the horizon, patients suffering from hematological malignancies may find themselves able to self-administer effective decitabine treatments from the comfort of their own home. As intravenous decitabine has already been identified as a cost-effective treatment of intermediate- to high-risk MDS, oral alternatives could offer an even more affordable option [167].

The molecular mechanism of nucleoside transporters in cell membranes responsible for the uptake of natural and synthetic nucleotides is a cardinal area of focus for future research. Cellular uptake is regulated by the following two transporter families: human equilibrative nucleoside transporters (hENTs) and human concentrative nucleoside transporters (hCNTs). Radiolabeled azacitidine in *Xenopus laevis* oocyte revealed the azanucleoside to be transported by all seven human nucleoside transporters (hNTs) (hENT1, hENT2, hENT3, hENT4, hCNT1, hCNT2, and hCNT3) with hCNT3 with the highest affinity for direct uptake of azacitidine [168]. Furthermore, gemcitabine, another nucleoside analogue, also displayed transportability through all hNTs with hCNT3 with the highest uptake while only hENT1 and hENT2 were able to transport the DNMTi, decitabine [168]. As such, the presence of the transmembrane hNT transporters is a necessity to induce the mechanism of cytotoxicity at pharmacologically relevant high concentrations [43,169]. The transportation of azacitidine by hENT2 demonstrated a crucial role in the indication of induced cytotoxicity due to the presence of the hNT in tumor cells [168].

This information can be used to deduce nucleoside drug potency levels and response since specific nucleoside transporters can serve as biomarkers for effective drug entry and subsequent action [43]. This will enable more accurate measurements of molecularly tracing drug molecules in cancer cells, determining the cellular uptake frequency for various azanucleosides. Moreover, the tissue-specific cellular expression of nucleoside transporters has been associated with cytotoxicity induced by the nucleoside analogs for either antiviral or anticancer therapeutics. The substrate specificity of the various nucleosides and analogs are explicit to the corresponding transmembrane protein transport system. Clinical and experimental trials investigating the expression patterns of the transporters and will lead to more development of nucleoside analogs for various diseases [169]. Novel studies comparing the cellular uptake and nucleic acid integration of both drugs, azacitidine and decitabine, need to be conducted to generate pharmacogenetics data for patients and clinicals in choosing a particular drug based on genomic profiling.

Azacitidine is an FDA approved treatment for patients with MDS, CMML, and AML with up to 30% blasts, and it is a preferred treatment option for patients who are ineligible for standard induction chemotherapy [112]. Clinical data of phase III trials investigating azacitidine solidifies the drug as an effective treatment option over standard/conventional care. Azacitidine is currently available in two formulations: vidaza, an injectable form, and CC-486, an oral form. Vidaza is administered either subcutaneously or intravenously initially at a primary treatment cycle of 75 mg/m^2^ daily for 7 days. Further treatment cycles can be administered upon monitoring hematological response, and the additional cycles can be repeated every 4 weeks with a maximum increase to 100 mg/m^2^. The oral formulation CC-486 was approved by the FDA as a maintenance therapy for newly diagnosed AML patients who responded to front-line chemotherapy due to the results of the QUAZAR AML-001 study. Oral azacitidine (CC-486) in combination with the best supportive care is currently being evaluated for efficacy and safety over the injectable azacitidine (ClinicalTrials.gov Identifier: NCT04806906) in a phase 2 pilot study. An oral formulation is clinically desired due to the opportunity of delivering at lower systemic doses over a more prolonged schedule. Moreover, the route of oral administration will alleviate the injection site reactions associated with vidaza.

Currently, there are no effective treatments to suppress leukemic transformation and increase survival outcomes in patients with higher-risk MDS. Allogenic haemopoietic stem-cell transplantation offer the only potentially curative treatment plan for this group of patients. As such, DNMTis such as azacitidine offer an alternative treatment that provides significant overall survival benefits for those with MDS. In a randomized controlled phase III (CALGB) trial, azacitidine treatment yielded a significantly higher response rate, improved quality of life, reduced risk of leukemic transformation, and increased survival compared to that of supportive care for high-risk MDS patients [93]. Conventional care for high-risk MDS patients or those categorized as intermediate-2 or high on IPSS involves the implementation of either supportive care, low-dose cytarabine, or intensive chemotherapy. In a subsequent phase III trial (ClinicalTrials.gov Identifier: NCT00071799), median overall survival for the azacitidine group was 24.5 months in comparison to the 15 months for the conventional care group [170].

Individual azacitidine treatment has proven efficacious, but current research is focused on combinations of histone deacetylase inhibitor (HDACi) drugs with azacitidine to increase the overall safety and effectiveness of treatments. HDACs act on specific histone protein tails and selectively alter gene transcription by changing chromatin status to regulate gene transcription [171]. The epigenetic combination of DNMT and HDAC inhibitors induces synergistic re-expression of silenced genes involved in modulating the cell cycle through individual drug mechanisms. Positive therapeutic reactions from the combination therapy consisting of azacitidine and venetoclax have already demonstrated that conjugation of specific inhibitors with DNMT inhibitors improved clinical outcomes. In a phase III (ClinicalTrials.gov Identifier: NCT02993523) trial, the median overall survival for the azacytidine-venetoclax group was 14.7 months compared to 9.6 months (hazard ratio for death, 0.66; *p* < 0.001) for the azacitidine-placebo control group for previously untreated patients with AML [99]. Further clinical studies (ClinicalTrials.gov Identifiers: NCT04401748, NCT01566695, etc.) (Table 1) of the combination of azacitidine and venetoclax are currently active to reinforce the significance of the positive outcomes to survival for late MDS and AML onsets. Studies are underway with specific HDACi like mocetinostat, valproic acid, pracinostat, and romedepsin combined with either hypomethylating agents or purine analogs to serve as therapeutics for hematological malignancies [172]. For instance, the phase 3, multicenter, double-blind, randomized PRUMULA study (ClinicalTrials.gov Identifier: NCT03151408) is evaluating the efficacy of the combination therapy between azacitidine and pracinostat in newly diagnosed AML patients who are ineligible to receive stem cell transplantation or chemotherapy [173]. As these studies progress, insights will reveal the viability of these substituted therapies for patients who are over 75 years old or have a protocol-defined comorbidity/other clinical complications.

Moving to proteasome inhibitors, recent clinical trials have already explored bortezomib’s apoptotic induction abilities in various types of cancer, including hematological malignancies [35,141,142,143]. Its targeted effect via reduced expression of genes associated with conditions like AML suggest its role as a treatment strategy in such malignancies, especially when used in combination with other chemotherapeutics [144,145]. Early in vivo studies show increased survival in mouse models of AML and hematological improvements in humans with MDS [40,145]. Phase III clinical trials are limited in their results and vary in their conclusions regarding bortezomib’s effect in hematological malignancies. A combined treatment regimen of bortezomib, melphalan, and prednisone in refractory MM patients resulted in reduced risk of death and increased TTNT [147]; however, a similar study in AML patients combined bortezomib with standard chemotherapies used to treat the condition and found no significant improvements when adding the drug to prescribed regimens [149]. Therefore, more laboratory research is needed along with phase III clinical trials to better our understanding of bortezomib’s role in the treatment of hematological malignancies other than MM.

Further understanding of bortezomib’s specific mechanism of action can help clarify its potential role in possibly slowing MDS/AML disease development and progression. Bortezomib’s proteasome binding abilities have been well recorded and its relationship to the NF-kB pathway has been thoroughly explored. Overactivation related mutations in NF-kB signaling molecules have been identified in the pathogenesis of MM [123]; overactive molecules in this pathway promote the growth of MM cells via expression of growth-promoting cytokines like IL-6, cell-cycle regulators that decrease cells’ sensitivity to cycle arrest, anti-apoptotic molecules, telomerase reverse transcriptase (TERT), angiogenic factors like VEGFs, and pro-adhesion molecules [123]. Therefore, bortezomib’s inhibitory effect on this pathway remains a reasonable target for inhibiting several cellular mechanisms that promote MM cell growth and survival. Additionally, bortezomib has been shown to revive autophagic mechanisms lost in malignancies like MDS. Autophagy-related genes (*atgs*), specifically Atg3, are downregulated in conditions like MDS; after overexpressing Atg3 in leukemic cell lines, researchers found malignant cells more sensitive to bortezomib-induced autophagy [174]. This highlights the drug’s effect in initiating ER overload, oxygen build-up, and protein dysregulation, leading to DNA damage and subsequent cell death [via autophagy] [34,117].

Researchers are still working to define mechanisms by which the drug influences transcript levels of cellular stress response genes and other genes related to malignancy. A study looking at transcription signatures in MDS and AML identified a relationship between NF-kB pathway inhibition and TNF receptor 6 (CD95/Fas) expression, an important component in initiating apoptosis [175]. Low-risk MDS patients display higher levels of the *Fas* receptor; however, when the disease progressed to AML the *FAS* gene was epigenetically repressed in 60% of patients [176]. Interestingly, methylation at NF-kB sites is related to *FAS* gene silencing [175]; given bortezomib’s inhibitory effect on the NF-kB pathway, this could be an important area for future research, especially considering its mixed success in phase III trials treating AML patients and limited data in MDS patients [148,149].

Recent research is also seeking to define concrete relationships between bortezomib and the epigenome, as the drug has been implicated in RNA interference pathways and DNA methylation regulation. Researchers have continued to explore bortezomib’s influence over the transcription of genes associated with DNA damage repair and other genotoxic stress responses, specifically in cases where malignant cells display resistance to the drug. A recent study found MM cells resistant to bortezomib showed significant downregulation of the SENP2 gene; this gene encodes for a serine-protease heavily involved (de)sumoylation [177]. Sumoylation and deSUMOylation are posttranslational protein modification processes crucial to gene regulation, DNA damage responses, signal transduction, and cell cycle control [177], and many of these pathway components are found dysregulated in cancers [178]. Moreover, this study uncovered a link between bortezomib resistance, SENP2 expression, and the NF-kB pathway; loss of SENP2 expression results in increased sumoylation of IkBα, and subsequent activation of NF-kB [177]. Recent research has also uncovered possible synergistic interactions between other epigenetic drugs and proteasome inhibitors like bortezomib and carfilzomib. Epigenetic proteins like HDAC6 are involved in the degradation of misfolded proteins via aggresomal degradation [179]. When used in-tandem with bortezomib and carfilzomib, inhibition of HDAC6 using HDAC6 selective inhibitor ricolinostat induced anti-MM effects in vitro and in vivo [179,180,181,182]. Carfilzomib was also found to work synergistically with phosophoinositidine 3-kinase (PI3K) inhibitor TGR-1202 to prevent c-Myc translation and c-Myc dependent transcription, contributing to lymphoma and leukemia cell death [183]. Therefore, combined use of epigenetic drugs and proteasome inhibitors like bortezomib and carfilzomib could provide avenues for potential novel treatments that target multiple protein degradation and transcriptional pathways related to hematological malignancies.

Given its role in the pathogenesis of hematological malignancies like AML, and its responsiveness to bortezomib, CEBPD could also be a reasonable target for increasing bortezomib’s effectiveness in vivo. As mentioned above, CEBPD is methylated and silenced in AML patients. Therefore, hypomethylating agents such as 5-Aza-deoxycytidine could offer a novel approach to bolster bortezomib’s effect in hematological malignancies, in combination with pre-existing treatments [135]. Given the bortezomib’s activation of CEBPD, examining pathways associated with CEPBD’s mechanism could also offer potential avenues for combating drug resistance. The direct substrate of p38 MAPK, MAPKAPK2 (MK2), positively regulates MM cell proliferation and drug resistance [184]. Inhibition of MK2 prevents MM cell proliferation and increased survival in mouse models, specifically when this was done in conjunction with standard MM treatments like bortezomib, doxorubicin, and dexamethasone [184]. However, more research is needed to better understand the interplay between p38 MAPK, CEBPD and bortezomib before they can be verified as a viable axis for developing new MDS and AML treatments. Similar to hypomethylating agents like decitabine and azacytidine, altered cellular metabolism could also play a role in bortezomib resistance. For instance the bortezomib resistance found in MM cells was associated with altered glucose metabolism showing higher activity of pentose phosphate and serine synthesis pathways [185]. Consequently, deprivation of metabolic intermediates essential to these glucose metabolic pathways offer a novel experimental platform for addressing bortezomib resistance towards the goal of preventing relapse and disease progression [185].

These findings are promising as researchers look to identify possible targets for eliminating bortezomib resistance in patients who have acquired or intrinsic resistance to the drug, and as more proteasome inhibitors are developed in order to overcome such resistances. Since the examination of proteasome inhibition as cancer treatments, researchers have also become interested in other aspects of protein ubiquitination as it relates to cancer development. Recent research has focused on deubiquitinating enzymes (DUBs) and how they can be selectively inhibited to create effective chemotherapeutics [186]. These pathways intersect heavily with the action of bortezomib as a proteasome inhibitor, and therefore could further elucidate bortezomib’s efficacy in combinatory chemotherapeutic treatment regimens in a wide variety of hematological malignancies and potentially solid tumors.

Collectively, decitabine, azacitidine and bortezomib can remodel the epigenome of cancer cells to prevent their proliferation and are currently used as chemotherapeutic treatments in the clinic. However, their specific mechanisms of action remain under investigation, and they are being used as a basic research tool to elucidate new epigenetic programs associated with MDS and its potential to transition into AML.

## Figures and Tables

**Figure 1 pharmaceuticals-14-00641-f001:**
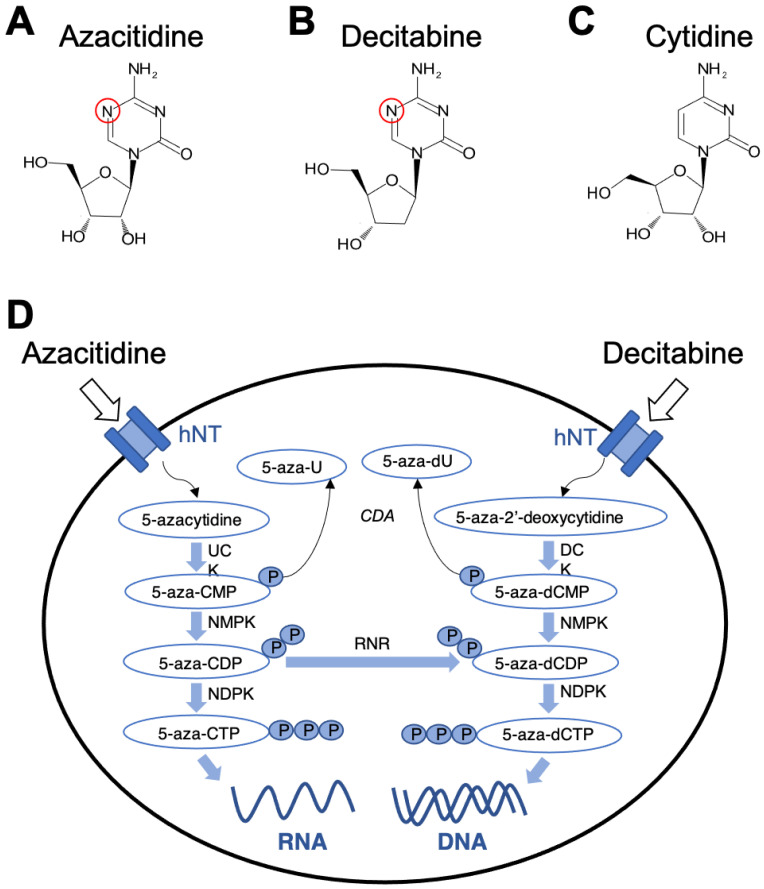
Incorporating mechanisms of azacytidine and decitabine into RNA and DNA. Chemical structures of (**A**) azacytidine, and (**B)** decitabine, which are analogs of (**C**) cytidine. (**D**) Both azacitidine and Decitabine have similar mechanisms of integration into RNA and DNA, respectively. These nucleoside analogs are transported into the cell via human nucleoside transport (hNT) channel. Azacitidine (5-azacytidine) becomes phosphorylated via the uridine cytidine kinase (UCK) to form 5-aza-CMP analog, which becomes successively phosphorylated via nucleoside monophosphate kinases (NMPK) and nucleoside diphosphatase kinase (NDPK) to create 5-aza-CDP and 5-aza-CTP, respectively. Decitabine (5-aza-2′-deoxycytodine) becomes phosphorylated by deoxycytidine kinase (DCK) to make 5-aza-dCMP analog, which becomes successively phosphorylated via NMPK and NDPK to form 5-aza-dCDP and 5-aza-dCTP, respectively. Once these ribonucleoside and deoxyribonucleoside analogs are in their active states (5-aza-CTP, 5-aza-dCTP), they can replace cytosines in RNA and DNA, respectively. Note that the monophosphate analogs, 5-aza-CMP and 5-aza-dCMP, can be deaminated into 5-aza-U and 5-aza-dU uridine analogs respectively, by cytidine deaminase (CDA), which helps maintain the surplus of pyrimidines. Additionally, the diphosphate analog 5-aza-CDP, can be converted into 5-aza-dCDP via ribonucleotide reductase (RNR) by reducing the ribonucleoside into a deoxyribonucleoside.

**Figure 2 pharmaceuticals-14-00641-f002:**
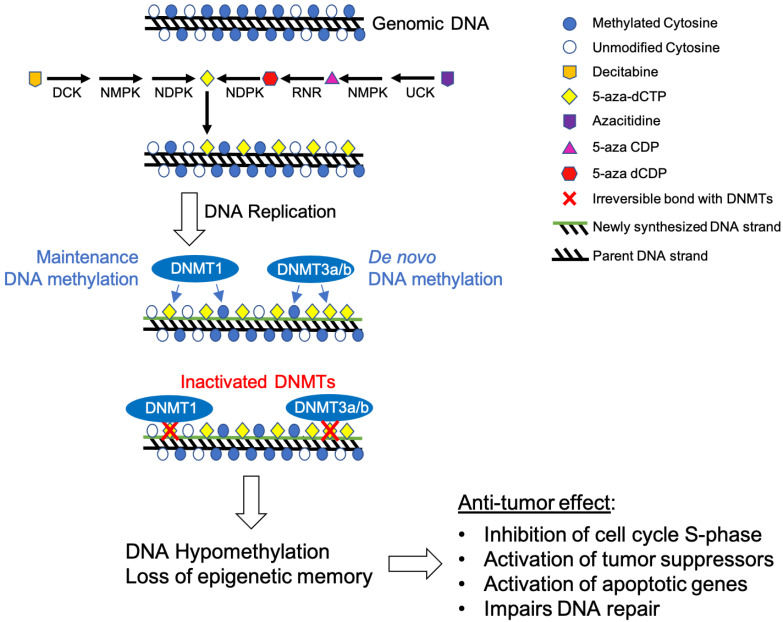
Inhibition of DNMTs upon incorporation of decitabine and azacytidine into DNA. The genomic DNA methylation landscape consists of methylated and unmethylated cytosines. Upon multiple phosphorylation steps by DCK (deoxycytidine kinase), NMPK (nucleoside monophosphate kinases) and NDPK (nucleoside diphosphatase kinase), decitabine becomes 5-aza-dCTP, which gets incorporated into DNA. Upon phosphorylation steps by UCK (uridine cytidine kinase) and NMPK (nucleoside monophosphate kinases), azacytidine is converted into 5-aza-CDP, which is recognized by ribonucleotide reductase (RNR) leading to the formation of 5-aza-dCDP. This reactive analog is further phosphorylated to 5-aza-dCTP by NDPK (nucleoside diphosphatase kinase), which is then integrated into genomic DNA. Upon DNA replication, DNMT1 maintains the methylation status of the genome. However, DNMT1 becomes covalently bound to the cytosine analog, 5-aza-dCTP, which prevents DNMT1 activity leading to genomic hypomethylation and loss of epigenetic memory. Additionally, other DNMT enzymes, DNMT3a and DNMT3b, can also form an irreversible interaction with azanucleosides leading to hypomethylation of the genome. Collectively, this drug-mediated DNA hypomethylation causes anti-tumorigenic effects including inhibition of the cell cycle, DNA repair impairments, activation of pro-apoptotic and tumor suppressor genes.

**Figure 3 pharmaceuticals-14-00641-f003:**
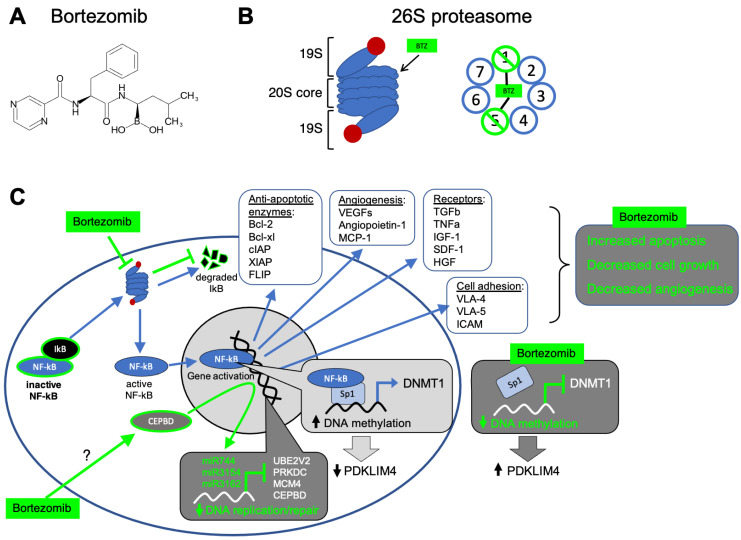
Cellular and epigenetic mechanisms of bortezomib in hematological malignancies. (**A**) Chemical structure of Bortezomib. (**B**) After entering the cell, bortezomib interacts with the 20S catalytic core of the proteasome, rendering the core’s β1 and β5 subunits inactive. (**C**) This proteosome inactivation interrupts the NF-kB pathway. In a malignant cell with unhindered proteasomal function, NF-kB’s inhibitory molecule, IkB, is degraded by the proteasome. Thus, in its active form, unbound NF-kB is free to enter the nucleus and activate transcription of genes that support survival and proliferation of malignant cells by upregulating the expression of cell adhesion molecules (i.e., VLA-4, VLA-5, ICAM), angiogenesis promoting factors (i.e., VEGFs, angioprotein-1, MCP1), anti-apoptotic enzymes (i.e., Bcl-2, Bcl-XL, cIAP, XIAP, FLIP), and various signaling molecules/receptors (i.e., IL-6, TGFβ, TNFα, IGF-1, SDF-1, HGF). Bortezomib-mediated loss of proteasomal function, keeps NF-kB bound to its inhibitor (IKB) and prevents its nuclear entry. Consequently, the inactive form of NF-kB—unable to enter the nucleus—disables Sp1-mediated transcriptional activation of DNMT1 gene. Under normal conditions, functional interaction between NF-kB and Sp1 activates transcription of DNMT1 gene, resulting in DNA hypermethylation. Bortezomib-mediated inhibition of NF-kB pathway decreases DNMT1 expression leading to hypomethylation of a tumor suppressor gene PDKLIM4 (PDZ and LIM domain 4). Mechanisms underlying bortezomib-mediated regulation of epigenetic pathways and gene targets are under investigation. By mechanisms yet to be determined, bortezomib can potentiate the transcription factor CCAAT/enhancer binding protein delta (CEBPD) leading to the upregulation of miRNAs (miR744, miR3154, miR3162) capable of repressing the expression of the oncogenes UBE2V2 (ubiquitin conjugating enzyme E2 V2), PRKDC (protein kinase DNA-activated catalytic subunit), and MCM4 (minichromosome maintenance complex component 4). Despite remaining mechanistic questions, bortezomib can block cellular pathways to facilitate malignant growth and alter DNA methylation, and thereby a treatment for hematological malignancies.

**Figure 4 pharmaceuticals-14-00641-f004:**
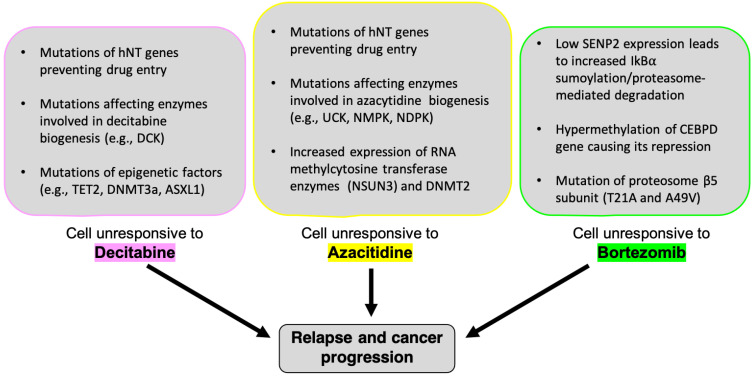
Mechanisms of drug resistance in hematological malignancies. The mechanisms of cellular resistance to decitabine and azacitidine are still under investigation, however, several studies have begun to elucidate processes associated with resistance to both drugs. Mutations in epigenetic proteins (i.e., DNMT1, DNMT3a, TET2, and ASXL1) are often associated with decitabine [and azacitidine] resistance; these genetic alterations result in a hindered DNA methylation process and interfere with both drug’s ability to induce hypomethylation. Recent research has uncovered connections between azacitidine resistance and altered chromatin structures, specifically as it relates to RNA methyltransferases (RCMTs). In azacitidine responsive cells, RCMTs NSUN3 and DNMT2 bind specific RNA binding proteins and transcription factors to form an azacitidine-sensitive chromatin structure. However, drug resistant MDS, AML, and leukemic cells display high levels of NSUN1-BRD4-RNA-Pol2 binding; this binding complex forms an active chromatin structure that is insensitive to azacitidine. Additionally, mutations in enzymes involved in the biogenesis of decitabine (i.e., deoxycytidine kinase [DCK], nucleoside monophosphate kinase [NMPK]) and azacitidine (i.e., nucleoside monophosphate kinase [NMPK], uridine cytidine kinase [UCK]) contribute to cellular resistances to these drugs, alongside mutations in pyrimidine metabolic pathways which alter cancer cells’ ability to effectively process both drugs. Finally, nucleoside uptake transporter mutations contribute to such resistances, as these mutations prevent decitabine/azacitidine entry and metabolic activation in malignant cells. Bortezomib resistances involve a variety of genetic alterations and cellular pathways. Mutations in and/or overexpression of the β5 subunit gene of the proteasome’s 20S catalytic core are known to be involved in bortezomib resistances. More specifically, two specific amino acid swaps (i.e., T21A and A49V) alter the binding pocket of the β5 subunit, decreasing bortezomib’s affinity for the subunit’s active site. Given its activation and epigenetic effects in myeloma cells treated with bortezomib, CCAAT/enhancer binding protein delta (CEBPD) may also have a role in conferring bortezomib resistant to malignant cells. Hypermethylation of the CEBPD gene is characteristic in many patients suffering from AML. CEBPD is also highly expressed in cells treated with drugs [like bortezomib] that activate the p38 MAPK pathway, suggesting a synergistic interaction between this pathway and bortezomib that results in CEBPD activation. Additionally, bortezomib resistant cells display low levels of SENP2, a serine protease involved in the sumoylation of IkBα, and subsequent activation of NF-kB. With a wide array of [interlocking] mechanisms associated with drug resistance, future research should focus on deconstructing these pathways’ involvement in cancer prognosis; doing so will help develop targeted treatment regimens that contribute to long-lasting remission and patient survival.

**Table 1 pharmaceuticals-14-00641-t001:** This table shows phase III clinical trials for decitabine, azacytidine and bortezomib in hematological malignancies.

Study Number	Status	Drug	Study Title	Phase	Masking	Indication/ Condition	No./Type of Participants	Start/Completion Dates
**NCT01809392**	Unknown	Decitabine	Decitabine Augments for Post Allogeneic Stem Cell Transplantation in Patients With Acute Myeloid Leukemia and Myelodysplastic Syndrome	2, 3	Open label, non-randomized	Acute Myeloid Leukemia, Myelodysplastic Syndromes	15	Jan 2013–Dec 2015
**NCT03377725**	Unknown	Decitabine, Arsenic Trioxide	Decitabine and Arsenic Trioxide for Myelodysplastic Syndrome (MDS)	3	Single, randomized	Myelodysplastic Syndromes, P53 mutation	300	March 2018–Nov 2020
**NCT02744742**	Unknown	Decitabine, Busulfan (BU), Cyclophosphamide (CY), Granulocyte Colony-Stimulating Factor(G-CSF)	G-CSF+Decitabine+BUCY vs. BUCY Conditioning Regimen for RAEB-1, REAB-2 and AML Secondary to MDS Undergoing Allo-HSCT	2, 3	Open label, randomized	Myelodysplastic Syndrome, Allogeneic Hematopoietic Stem Cell Transplantation, Conditioning	122	March 2019–March 2020
**NCT02214407**	Active, not recruiting	Decitabine, Hydroxyurea	Randomized Phase III Study of Decitabine +/− Hydroxyurea (HY) Versus HY in Advanced Proliferative CMML (GFM-DAC-CMML)	3	Open label, randomized	Myelodysplastic Syndrome	168	Oct 2014–Oct 2021
**NCT02272478**	Recruiting	Arm A: Mylotarg plus DA Versus CPX-351, Arm B: Vosaroxin and Decitabine, Arm D: Small molecule or Not, Arm C: DA V FLAG-Ida V DAC, Arm E: CPX-351 (200 V 300), Arm F: DA V IDAC	Trial to Test the Effects of Adding 1 of 2 New Treatment Agents to Commonly Used Chemotherapy Combinations (AML18)	2, 3	Open label, randomized	Acute Myeloid Leukemia, Myelodysplastic Syndrome	1600	Feb 2021–Feb 2022
**NCT03306264**	Recruiting	ASTX727, Dacogen (decitabine)	Study of ASTX727 vs. IV Decitabine in MDS, CMML, and AML	3	Open label, randomized	Myelodysplastic Syndromes, Chronic Myelomonocytic Leukemia, Acute Myeloid Leukemia	200	Feb 2018–May 2022
**NCT02172872**	Active, not recruiting	Standard combination chemotherapy, decitabine	“InDACtion” vs. “3 + 7” Induction in AML	3	Open label, randomized	Acute Myeloid Leukemia	606	Nov 2014–Dec 2022
**NCT04713956**	Recruiting	Granulocyte Colony-Stimulating Factor(G-CSF), Decitabine (DAC), Busulfan (BU), Cyclophosphamide (CY), Fludarabine (FLU)	G-CSF+DAC+BUCY vs. G-CSF+DAC+BF Conditioning Regimen for RAEB-1,REAB-2 and AML Secondary to MDS Undergoing Allo-HSCT	2, 3	Open label, randomized	Myelodysplastic Syndrome, Allogeneic Hematopoietic Stem Cell Transplantation, Conditioning	242	Jan 2021–July 2024
**NCT02085408**	Active, not recruiting	Clofarabine, daunorubicin hydrochloride, cytarabine, decitabine	Clofarabine or Daunorubicin Hydrochloride and Cytarabine Followed By Decitabine or Observation in Treating Older Patients With Newly Diagnosed Acute Myeloid Leukemia	3	Open label, randomized	Acute Myeloid Leukemia	727	Dec 2010–Oct 2024
**NCT04173533**	Recruiting	Oral Azacitidine, Matched placebo	Randomized Study of Oral Azacitidine vs. Placebo Maintenance in AML or MDS Patients After Allo-SCT (AMADEUS)	III	Double, Randomized	AML, Myelodysplasia	324	Jun 2019–Jun 2024
**NCT03092674**	Active, not recruiting	Azacitidine, Cytarabine, Decitabine, Midostaurin	Azacitidine With or Without Nivolumab or Midostaurin, or Decitabine and Cytarabine Alone in Treating Older Patients With Newly Diagnosed Acute Myeloid Leukemia or High-Risk Myelodysplastic Syndrome	II, III	Open Label, Randomized	AML, Myelodysplastic Syndrome	1670	Dec 2017–Aug 2023
**NCT03268954**	Recruiting	Azacitidine, Pevonedistat	Pevonedistat Plus Azacitidine Versus Single-Agent Azacitidine as First-Line Treatment for Participants With Higher-Risk Myelodysplastic Syndromes (HR MDS), Chronic Myelomonocytic Leukemia (CMML), or Low-Blast Acute Myelogenous Leukemia (AML) (PANTHER)	III	Open Label, Randomized	Myelodysplastic Syndrome (Leukemia, Myelonocytic, Chronic Leukemia, Myeloid, Acute)	502	Nov 2017–Feb 2025
**NCT00071799**	Completed	Azacitidine	A Survival Study in Patients With High Risk Myelodysplastic Syndromes Comparing Azacitidine Versus Conventional Care	III	Open Label, Randomized	Myelodysplastic Syndromes	358	Nov 2003–July 2007
**NCT04256317**	Recruiting	ASTX030 (Cedazuridine + Azacitidine)	A Study of ASTX030 (Cedazuridine in Combination With Azacitidine) in MDS, CMML, or AML	II, III	Open Label, Randomized	MDS, CMML, AML	245	May 2020–Apr 2023
**NCT03978364**	Recruiting	Azacitidine combined HHT, Azacitidine regimen	A Study of Azacitidine for Patients With Int/High -Risk MDS and AML-MRC	III	Open Label, Randomized	Myelodysplastic Syndrome, AML	100	Jun 2019–Dec 2022
**NCT00887068**	Completed	Azacitidine	Controlled Study of Post-transplant Azacitidine for Prevention of Acute Myelogenous Leukemia and Myelodysplastic Syndrome Relapse (VZ-AML-PI-0129)	III	Open Label, Randomized	AML, MDS	187	Apr 2009–Aug 2018
**NCT03173248**	Recruiting	AG-120 (ivosidenib) with Azacitidine, Placebo with Azacitidine	Study of AG-120 (Ivosidenib) vs. Placebo in Combination With Azacitidine in Patients With Previously Untreated Acute Myeloid Leukemia With an IDH1 Mutation (AGILE)	III	Triple. Randomized	AML (Newly Diagnosed, Untreated AML, AML arising from MDS, Leukemia, Myeloid, Acute	200	Jun 2017–Jun 2022
**NCT04842604**	New, not yet recruiting	Glasdegib, Azacitidine	Continuation Study of B1371019(NCT03416179) and B1371012(NCT02367456) Evaluating Azacitidine With Or Without Glasdegib In Patients With Previously Untreated AML, MDS or CMML	III	Open Label, Non-Randomized	AML, MDS, CMML	37	Apr 2021–Dec 2022
**NCT04401748**	Recruiting	Venetoclax, Azacitidine, Placebo	Study Of Venetoclax Tablet With Intravenous or Subcutaneous Azacitidine to Assess Change in Disease Activity In Adult Participants With Newly Diagnosed Higher-Risk Myelodysplastic Syndrome (Verona)	III	Quadruple, Randomized	MDS	500	Sep 2020–Feb 2025
**NCT01566695**	Active, not recruiting	Venetoclax, Azacitidine, Placebo	The Efficacy and Safety of Oral Azacitidine Plus Best Supportive Care Versus Placebo and Best Supportive Care in Subjects With Red Blood Cell (RBC) Transfusion-Dependent Anemia and Thrombocytopenia Due to International Prognostic Scoring System (IPSS) Low Risk Myelodysplastic Syndrome (MDS)	III	Quadruple, Randomized	Myelodysplastic Syndrome	216	Apr 2013–Dec 2021
**NCT04313881**	Recruiting	Magrolimab, Azacitidine, Placebo	Magrolimab + Azacitidine Versus Azacitidine + Placebo in Untreated Participants With Myelodysplastic Syndrome (MDS) (ENHANCE)	III	Double, Randomized	Myelodysplastic Syndrome	520	Sep 2020–Aug 2025
**NCT03416179**	Active, not recruiting	Glasdegib, Daunorubicin + Cytarabine, Azacitidine, Placebo	A Study Evaluating Intensive Chemotherapy With or Without Glasdegib or Azacitidine With or Without Glasdegib In Patients With Previously Untreated Acute Myeloid Leukemia (BRIGHT AML1019)	III	Quadruple, Randomized	Untreated AML, Leukemia, Myeloid, Acute	731	Apr 2018–Dec 2022
**NCT01109004**	Completed	Lenalidomide, bortezomib, dexamethasone	Stem Cell Transplant With Lenalidomide Maintenance in Patients With Multiple Myeloma (BMT CTN 0702)	3	Open label, randomized	Multiple Myeloma	758	May 2010–March 2018
**NCT02811978**	Completed	Bortezomib (Velcade), Dexamethasone	Study of Subcutaneous and Intravenous Velcade in Combination With Dexamethasone in Chinese Subjects With Relapsed and Refractory Multiple Myeloma	3	Open label, randomized	Multiple Myeloma	81	Sept 2016–Nov 2018
**NCT01146834**	Completed	Bortezomib (Velcade), cyclophosphamide, G-CSF, Plerixafor	Trial of Three Stem Cell Mobilization Regimens for Multiple Myeloma	3	Open label, randomized	Multiple Myeloma	47	March 2011–Feb 2019
**NCT02112916**	Active, not recruiting	Bortezomib	Combination Chemotherapy With or Without Bortezomib in Treating Younger Patients With Newly Diagnosed T-Cell Acute Lymphoblastic Leukemia or Stage II-IV T-Cell Lymphoblastic Lymphoma	3	Open label, randomized	T-Cell Acute Lymphoblastic Leukemia, Stage II-IV T-Cell Lymphoblastic Lymphoma	844	Sept 2014–March 2020
**NCT02195479**	Active, not recruiting	Velcade, Melphalan, Prednisone, Daratumumab IV and SC, Dexamethasone	A Study of Combination of Daratumumab and Velcade (Bortezomib) Melphalan-Prednisone (DVMP) Compared to Velcade Melphalan-Prednisone (VMP) in Participants With Previously Untreated Multiple Myeloma	3	Open label, randomized	Multiple myeloma	706	Dec 2014–Oct 2021
**NCT02136134**	Active, not recruiting	Bortezomib, Daratumumab	Addition of Daratumumab to Combination of Bortezomib and Dexamethasone in Participants With Relapsed or Refractory Multiple Myeloma	3	Open label, randomized	Multiple Myeloma	499	Jan 2014–Sept 2021
**NCT01208662**	Active, not recruiting	Lenalidomide, Bortezomib, Dexamethasone	Randomized Trial of Lenalidomide, Bortezomib, Dexamethasone vs. High-Dose Treatment With SCT in MM Patients up to Age 65 (DFCI 10-106)	3	Open label, randomized	Multiple Myeloma	660	Sept 2010–Sept 2023
**NCT03110562**	Active, not recruiting	Bortezomib, Selinexor, Dexamethasone	Bortezomib, Selinexor, and Dexamethasone in Patients with Multiple Myeloma (BOSTON)	3	Open label, randomized	Multiple Myeloma	402	May 2017–Sept 2023
**NCT01371981**	Active, not recruiting	Bortezomib, Sorafenib Tosylate	Bortezomib and Sorafenib Tosylate in Treating Patients With Newly Diagnosed Acute Myeloid Leukemia	3	Open label, randomized	Acute Myeloid Leukemia	1645	June 2011–Sept 2027

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
