# Peer review of "Hypomethylating Chemotherapeutic Agents as Therapy for Myelodysplastic Syndromes and Prevention of Acute Myeloid Leukemia"

_pharmaceuticals, 2021, doi:10.3390/ph14070641_

Round 1

Reviewer 1 Report

Well written review article about treatment of MDS with hypomethylating agents and prevention of progression to AML.

Abstract: As written now, it looks like bortezomib is an established treatment in MDS. In addition, the sentence “Phase III clinical trials….”  is misleading, because there are no large studies about bortezomib in MDS establishing its role in this disease. Therefore, the abstract needs to be rewritten to clarify phase III data on HMA and novel developments like bortezomib.

Introduction: Line 38/39: percentage should be behind blast count? Please omit or clarify %: differences between PB and BM. Dysplasia is as well needed and RSB as well as Auer rods are of importance.

Page 3 line 134 area

Page 4 line 178 SLC22AF should be SLC22A4

Page 5 line 209: Effect of decitabine on myeloma cells. Please check references 53/54

Page 7 line 345 CDKN2B should be A

Page 8 phrase line 389 “However…” please clarify what you mean. If no mechanism for AZA is found in MDS, why does it work?

Page 8 line 387. Citation of the Almasri paper is messing.

Page 8 line 397. Please no brand names or explain (ONUREG).

Page 9 line 433/447, Grade 3/4 not ¾

Page 9: side effect. Skin problems due to sc application. Please ad a sentence and how you manage.

Page 9: add some remarks to guadecitabine and where this drug could improve outcome compared to AZA/DEC.

Page 13 line 634 referencing the castor study does show, that Bor works as well in combination. However, the additional effect is due to Dara. I would rather cite a study which shows, that the addition Bor adds an effect to an other back bone.

Is there evidence of a group effect of the protesome inhibitor in MDS cells? Carfilzomib? Ixazomib?

Conclusions:

The first conclusion is hard to understand from a clinical standpoint. “Decitiabine is the most effective drug in MDS, AML and related cancers.” In the upper part, you outline the equal efficacy of Dec/Aza. In the following page, you outline once again the cellular effect of Dec and its potential combination. There is not much about Aza and it looks like Aza is only effective in combinations which is not true. In the following, the oral formulation of Dec is outlined while the one with AZA which is already approved in AML as maintenance is not mentioned.

P17/line 820. Bortezomib looks like an equal drug in MDS and AML which is not the case. In contrary, there was no clinical benefit in the randomized trial (even more harm). I agree, that there are convincing effects on the cellular basis, which make Bor an attractive new option in MDS but neither the single drug dose nore a dose in drug combination is clear until know and “promising results in reviving autophagic mechanisms lost in malignancies like MDS” is hypothesis generating but by far not a proof of principle. If sometimes the term “is effective in hematologic disease” is used and then conclusion are drawn for AML/MDS, this needs to be done with caution. As you outline later in the text: Bor is a very effective treatment in MM, there is no clear standing in MDS/AML. This should be the first statement and the than you should explain your hypothesis, why this will change in the future.

The paragraph starting from line 875 outlines the way, how the conclusions should be written if the main focus is on AZA/Dec/Bor. Bortezomib as a promising but not established drug in MDS can be combined with established drugs like Dec/Aza because of synergistic mechanism in cellular models. An other option would be rewriting the article in a broader way and list mechanisms of action on the epigenome in MDS and how all these drugs could be combined. This could then be clarified including the running studies listed in table 1 (rational behind these studies, why which dose of each drug, why one drug should be given prior to other drugs ...-> based on in vitro studies). Therefore, a  figure should sum up all ways of actions of different drugs on epigenome.

Author Response

Reviewer 1:

 Abstract: As written now, it looks like bortezomib is an established treatment in MDS. In addition, the sentence “Phase III clinical trials….”  is misleading, because there are no large studies about bortezomib in MDS establishing its role in this disease. Therefore, the abstract needs to be rewritten to clarify phase III data on HMA and novel developments like bortezomib.

Response:

  • We thank the reviewer for this insightful comment. The abstract has been reworded to clarify that phase III trials of decitabine and azacytidine only demonstrate their ability to be effective in treating MDS and AML patients. We then further clarified that phase III clinical trials of bortezomib treatments in MDS/AML patients are limited; however, the mechanism of action for bortezomib in multiple myeloma (MM), as well as various in vitro and in vivo studies, demonstrate that this drug has potential for treating MDS/AML patients. However, for this reason, we avoid drawing parallels between all three drugs throughout the manuscript.

Introduction: Line 38/39: percentage should be behind blast count? Please omit or clarify %: differences between PB and BM. Dysplasia is as well needed and RSB as well as Auer rods are of importance.

Response:

  • We clarified the various criteria used to diagnose MDS and AML, from general symptoms to cytologically specific observations. Cytopenias resulting in infection and bleeding susceptibility are mentioned first. Then, dysplasias of bone marrow and peripheral blood are mentioned. MDS subtypes are mentioned, with specific focus on how both cytopenia type and dysplasia type characterize each subtype. RSBs are mentioned, followed by a clarification regarding bone marrow blast count, and which percentages are associated with poor prognosis. Finally, Auer bodies/rods are mentioned and described as a cytological characteristic of AML that can be clinically helpful in determining disease severity.

Page 3 line 134 area

We addressed this point.

Page 4 line 178 SLC22AF should be SLC22A4

We addressed this point.

Page 5 line 209: Effect of decitabine on myeloma cells. Please check references 53/54 

Response:

  • References 53/54 refer to decitabine’s effect in MDS patients, not multiple myeloma (MM) patients. Thus, we changed ‘multiple myeloma’ to ‘MDS.’

Page 7 line 345 CDKN2B should be A

We addressed this point.

Page 8 phrase line 389 “However…” please clarify what you mean. If no mechanism for AZA is found in MDS, why does it work?

Response:

  • We thank the reviewer for this comment and acknowledged the ambiguity associated with the sentence beginning with ‘however.’ The intention of the sentence was to highlight the need for further research to better understand the mechanisms for azacitidine. We deleted the sentence to eliminate any misconceptions. The table shows the use of azacytidine in Phase III trials. The underlying mechanisms of azacitidine were displayed in the figures and explained in the Azacitidine section.

Page 8 line 387. Citation of the Almasri paper is missing.

We addressed this point. Relevant citation is incorporated into the References.

Page 8 line 397. Please no brand names or explain (ONUREG).

We addressed this point. ONUREG was deleted, and the oral formulation name CC-486 was used instead.

Page 9 line 433/447, Grade 3/4 not ¾.

How we’ve addressed this: Change completed

Page 9: side effect. Skin problems due to sc application. Please ad a sentence and how you manage.

Response:

  • We elaborated more on the side effects of the application of the injectable form of azacitidine. Also, we addressed the management of complications caused by the injection site reaction. Adding this information is relevant for better understanding of Vidaza effect. Moreover, it allowed for a transition on the comparison between Vidaza and CC-486 in the discussion.

Page 9: add some remarks to guadecitabine and where this drug could improve outcome compared to AZA/DEC.

Response:

  • We added a new paragraph in the discussion on second-generation DNA methyltransferase inhibitor guadecitabine. In the discussion, updates on the timeline of the drug were stated along with latest reports on guadecitabine. An overall conclusive statement was included in comparing guadectibine to its parental HMA treatments for patients with hematological malignancies.

Page 13 line 634 referencing the castor study does show, that Bor works as well in combination. However, the additional effect is due to Dara. I would rather cite a study which shows, that the addition Bor adds an effect to an other back bone.

Response:

  • We’ve removed the specific mention of Dara clinical trial and instead discuss a phase III clinical trial in which bortezomib is added to a melphalan/prednisone backbone. In this trial, VMP (bortezomib group) resulted in a marked difference in reducing death risk and OS [for particular patient’s subgroups].

Is there evidence of a group effect of the protesome inhibitor in MDS cells? Carfilzomib? Ixazomib?

Response:

  • Although we could not find a defined group effect of proteasome inhibitors used in combination, we did add a brief discussion of how proteasome inhibitors such as bortezomib and carfilzomib could work in conjunction with more concretely defined epigenetic drugs to induce cell death in hematological malignancies such as MM and leukemia. Data on MDS/AML with respect to this combined effect is limited, but again, we are hoping the promise in MM studies will promote investigation of bortezomib, carfilzomib, and ixazomib being used in combination with epigenetic drugs in MDS/AML specifically

Conclusions:

The first conclusion is hard to understand from a clinical standpoint. “Decitiabine is the most effective drug in MDS, AML and related cancers.” In the upper part, you outline the equal efficacy of Dec/Aza. In the following page, you outline once again the cellular effect of Dec and its potential combination. There is not much about Aza and it looks like Aza is only effective in combinations which is not true. In the following, the oral formulation of Dec is outlined while the one with AZA which is already approved in AML as maintenance is not mentioned.

Response:

  • We thank the reviewer for this insightful comment. A more conclusive statement regarding azacitdine was incorporated into the discussion section. For instance, crucial points regarding this drug were restated, along with its oral formulation. More information on CC-486 can be found in the azacitidine section of the review paper. The effectiveness of azacitidine alone is emphasized, but it is also stated that combinations are being investigated to further increase efficacy of the treatment while reducing side-effects. As the reviewer suggested, it was necessary to distinguish azacitidine as a treatment alone, without being incorporated into drug combinations.

P17/line 820. Bortezomib looks like an equal drug in MDS and AML which is not the case. In contrary, there was no clinical benefit in the randomized trial (even more harm). I agree, that there are convincing effects on the cellular basis, which make Bor an attractive new option in MDS but neither the single drug dose nore a dose in drug combination is clear until know and “promising results in reviving autophagic mechanisms lost in malignancies like MDS” is hypothesis generating but by far not a proof of principle. If sometimes the term “is effective in hematologic disease” is used and then conclusion are drawn for AML/MDS, this needs to be done with caution. As you outline later in the text: Bor is a very effective treatment in MM, there is no clear standing in MDS/AML. This should be the first statement and the than you should explain your hypothesis, why this will change in the future.

The paragraph starting from line 875 outlines the way, how the conclusions should be written if the main focus is on AZA/Dec/Bor. Bortezomib as a promising but not established drug in MDS can be combined with established drugs like Dec/Aza because of synergistic mechanism in cellular models. An other option would be rewriting the article in a broader way and list mechanisms of action on the epigenome in MDS and how all these drugs could be combined. This could then be clarified including the running studies listed in table 1 (rational behind these studies, why which dose of each drug, why one drug should be given prior to other drugs ...-> based on in vitro studies). Therefore, a figure should sum up all ways of actions of different drugs on epigenome.

Response:

  • We greatly appreciate these comments by the reviewer. Our abstract inaccurately framed bortezomib as an equally studied - along with azacitidine and decitabine - and effective drug in the treatment of MDS and AML. Thus, we reframed both abstract and discussion sections, rather than a total rewriting of the review, to address this issue. Upon reading the abstract, the reader is now aware that phase III clinical trial data related to bortezomib’s effect in MDS/AML is limited. With respect to bortezomib, the goal of this review is to provide both clinicians and research scientists with mechanistic information about the drug, specifically in multiple myeloma (MM), where it is best studied. From that mechanistic information, alongside experiments that highlight bortezomib’s specific effect on pathways related to MDS/AML development and progression, we hope to highlight how specific aspects of this mechanism could be further studied in order to more safely and effectively incorporate proteasome inhibitors like bortezomib into treatments for hematological malignancies such as MDS and AML. Moreover, we also highlight the need to elucidate the epigenetic effects of bortezomib and how these effects could be utilized synergistically with more concretely accepted/defined ‘epigenetic drugs’ such as decitabine and azacitidine, which are currently used in MDS/AML.

Reviewer 2 Report

In this manuscript, the authors review the role, mechanisms of action and known therapeutic effects (among clinical trials) of DNA hypo-methylation agent in the context of hematological malignancies. They focus on decitabine, azacitidine, and on the proteasome inhibitor bortezomib, more specifically in the context of myelodysplasia and acute myeloid leukemia.

The quality of the manuscript is excellent, of significant interest and relevant to myeloid malignancies research. The rational is clear; the manuscript is perfectly written and comprehensive. The discussion is clear and comprehensive.

Manuscript status; Accepted after Minor Revisions

Visible on the pdf but might not be eventually

Line 83:  space typos- leading to its _ use as a chemo  

Line 154: space typos - cellularly _by nucleoside transport proteins, including the equilibrative uniporters  

Line 357: typos - inhibiotor instead of inhibitor

Line 357: typos -  Investigatons instead of  Investigations

Author Response

Reviewer 2:

Manuscript status; Accepted after Minor Revisions

We kindly thank the reviewer for suggesting Acceptance of our review article with minor revisions.

Line 83:  space typos- leading to its _ use as a chemo 

We addressed this point.

Line 154: space typos - cellularly _by nucleoside transport proteins, including the equilibrative uniporters 

We addressed this point.

Line 357: typos - inhibiotor instead of inhibitor

We addressed this point.

Line 357: typos - Investigatons instead of Investigations

We addressed this point.

Round 2

Reviewer 1 Report

The revised version does incorporate the adressed points.